# Composition and structure of synaptic ectosomes exporting antigen receptor linked to functional CD40 ligand from helper T cells

David G Saliba[1,6†‡], Pablo F Céspedes-Donoso[1†], Štefan Bálint[1†],
Ewoud B Compeer[1], Kseniya Korobchevskaya[1], Salvatore Valvo[1], Viveka Mayya[1],
Audun Kvalvaag[1], Yanchun Peng[2,3], Tao Dong[2,3], Maria-Laura Tognoli[4],
Eric O'Neill[4], Sarah Bonham[5], Roman Fischer[5], Benedikt M Kessler[5],
Michael L Dustin[1]*

[1]Kennedy Institute of Rheumatology, Nuffield Department of Orthopaedics, Rheumatology and Musculoskeletal Sciences, University of Oxford, Oxford, United Kingdom; [2]MRC Human Immunology Unit, Weatherall Institute of Molecular Medicine, University of Oxford, Oxford, United Kingdom; [3]Nuffield Department of Medicine, Chinese Academy of Medical Science Oxford Institute, University of Oxford, Oxford, United Kingdom; [4]Department of Oncology, University of Oxford, Oxford, United Kingdom; [5]Discovery Proteomics Facility, Target Discovery Institute, Nuffield Department of Medicine, University of Oxford, Oxford, United Kingdom; [6]Department of Applied Biomedical Science, Faculty of Health Science, University of Malta, Msida, Malta

*For correspondence:
michael.dustin@kennedy.ox.ac.uk

†These authors contributed equally to this work

Present address: ‡Centre for Molecular Medicine and Biobanking, University of Malta, Msida, Malta

**Abstract** Planar supported lipid bilayers (PSLB) presenting T cell receptor (TCR) ligands and ICAM-1 induce budding of extracellular microvesicles enriched in functional TCR, defined here as synaptic ectosomes (SE), from helper T cells. SE bind peptide-MHC directly exporting TCR into the synaptic cleft, but incorporation of other effectors is unknown. Here, we utilized bead supported lipid bilayers (BSLB) to capture SE from single immunological synapses (IS), determined SE composition by immunofluorescence flow cytometry and enriched SE for proteomic analysis by particle sorting. We demonstrate selective enrichment of CD40L and ICOS in SE in response to addition of CD40 and ICOSL, respectively, to SLB presenting TCR ligands and ICAM-1. SE are enriched in tetraspanins, BST-2, TCR signaling and ESCRT proteins. Super-resolution microscopy demonstrated that CD40L is present in microclusters within CD81 defined SE that are spatially segregated from TCR/ICOS/BST-2. CD40L[+] SE retain the capacity to induce dendritic cell maturation and cytokine production.
DOI: https://doi.org/10.7554/eLife.47528.001

## Introduction

Immune response communication depends on intercellular interactions of surface receptors expressed on T cells and antigen presenting cells (APC) *via* immunological synapses (IS), kinapses or stabilized microvilli (*Cai et al., 2017*; *Mayya et al., 2018*). In model IS, receptor-ligand pairs organize into radially symmetric supramolecular activation clusters (SMACs). The central (c)SMAC incorporates a secretory synaptic cleft, TCR interaction with peptide-major histocompatibility complex (pMHC) and costimulatory receptor-ligand interactions and is surrounded by the peripheral (p)SMAC

enriched in LFA-1 (T cell side) interaction with ICAM-1 (APC side) enriched peripheral (p)SMAC (*Monks et al., 1998*). The dynamics of IS formation involves initial contacts through microvilli that trigger cytoplasmic $Ca^{2+}$ elevation leading to rapid spreading and formation of SMACs through inward directed cytoskeletal transport (*Grakoui et al., 1999*; *Kaizuka et al., 2007*). Once the IS matures, TCR-pMHC pairs form in the distal (d)SMAC and segregate into microclusters (MCs) that integrate signaling as they centripetally migrate to the cSMAC where signaling is terminated (*Vardhana et al., 2010*). TCR MCs are a common feature of IS, kinapses and stabilized microvilli (*Cai et al., 2017*; *Kumari et al., 2015*). However, the IS is not only a platform for signal integration, but also enables polarized delivery of effector function. These include the polarized delivery of cytokines (*Huse et al., 2006*), nucleic acid containing exosomes (*Mittelbrunn et al., 2011*), and TCR enriched extracellular vesicles that bud directly into the synaptic cleft from the T cell side of the IS (*Choudhuri et al., 2014*). 'Ectosomes' (also called microvesicles) are extracellular vesicles released from the plasma membrane (*Hess et al., 1999*). Therefore, we define TCR enriched extracellular vesicles that are formed in and simultaneously exported across the IS as synaptic ectosomes (SE).

CD40 ligand (CD40L, CD154) is a 39 kDa glycoprotein expressed by $CD4^+$ T cells (*Noelle et al., 1992*) and is one of the key effectors delivered by helper T cells through the IS (*Ridge et al., 1998*; *Schoenberger et al., 1998*). Inducible T cell costimulator (ICOS, also known at CD278) interaction with ICOSL promotes CD40L-CD40 interactions in the IS (*Liu et al., 2015*; *Papa et al., 2017*). CD40L is transferred to antigen presenting cells in vitro (*Gardell and Parker, 2017*). Trimeric CD40L released by proteolysis by ADAM10 is a partial agonist of CD40, suggesting the fully active CD40 must remain membrane anchored to sufficiently crosslink CD40 for full agonist function (*Yacoub et al., 2013*; *Haswell et al., 2001*). How helper T cells achieve this high level of crosslinking in the IS is not established.

In this study we set out to determine the protein composition and mechanism of SE release in the synaptic cleft by helper T cells. To this aim we develop technologies for isolation of SE released by T cells directly at the IS on BSLB (*Baksh et al., 2004*) and integrate complementary flow cytometry, mass spectrometry and super resolution microscopy data. We show that the polarized transfer of T cell derived SE is determined by selective sorting processes directly in the IS and depends on both the presence of ligands on the SLB and their segregation into the synaptic cleft, as shown for TCR complex:anti-CD3ε/pHLA-DR complexes, CD40L:CD40 and ICOS:ICOSL, but not LFA-1:ICAM-1 bound pairs. Other components, such as tetraspanins and BST-2, are enriched in SE without being engaged with a ligand. Quantitative mass spectrometry of SE revealed members of the core ESCRT machinery and adaptor proteins responsible for the scission of SE at the IS. Using direct stochastic optical reconstruction microscopy (dSTORM) we further demonstrate that individual SE often contain discrete TCR/ICOS/BST-2 and CD40L microclusters. SE budding in the IS, therefore, provides a strategy to generate antigen specific and effector armed structures that are freed from the T cell.

## Results

### CD40L is recruited to the IS and left by kinapses in a CD40 dependent manner

CD40L is stored in intracellular compartments within $CD4^+$ effector cells and mobilized to IS where it engages CD40 (*Koguchi et al., 2007*; *Boisvert et al., 2004*). To mimic the APC surface and stimulate IS formation, the PSLB presented the adhesion molecule ICAM-1 and a Fab fragment of the anti-CD3ε mAb UCHT1 (UCHT1-Fab) (*Choudhuri et al., 2014*), which functions like a strong agonist pMHC (*Schubert et al., 2012*) (*Figure 1A*). Due to challenges with fluorescent protein tagging of CD40L, we detected it in the IS using an anti-CD40L mAb, which has the caveat that it competes with CD40, but nonetheless detects recruitment of CD40L to the IS (*Papa et al., 2017*). To determine the impact of CD40 density in the PSLB on detection of CD40L by this method we allowed IS to form on PSLBs presenting ICAM-1 and UCHT1-Fab over the physiological range of CD40 densities from 0 to 500 molec./μm². The anti-CD40L signal was imaged by total internal reflection fluorescence microscopy (TIRFM) that only illuminates up to 200 nm into the sample, and thus restricts detection to the IS. Minimal IS CD40L was detected in the absence of CD40 as previously reported (*Papa et al., 2017*) and near uniformly increased anti-CD40L was detected at 10, 50 and 100 CD40 molec./μm² with a reduction in signal at 500 CD40 molec./μm² (*Figure 1B*). Thus, whether this loss

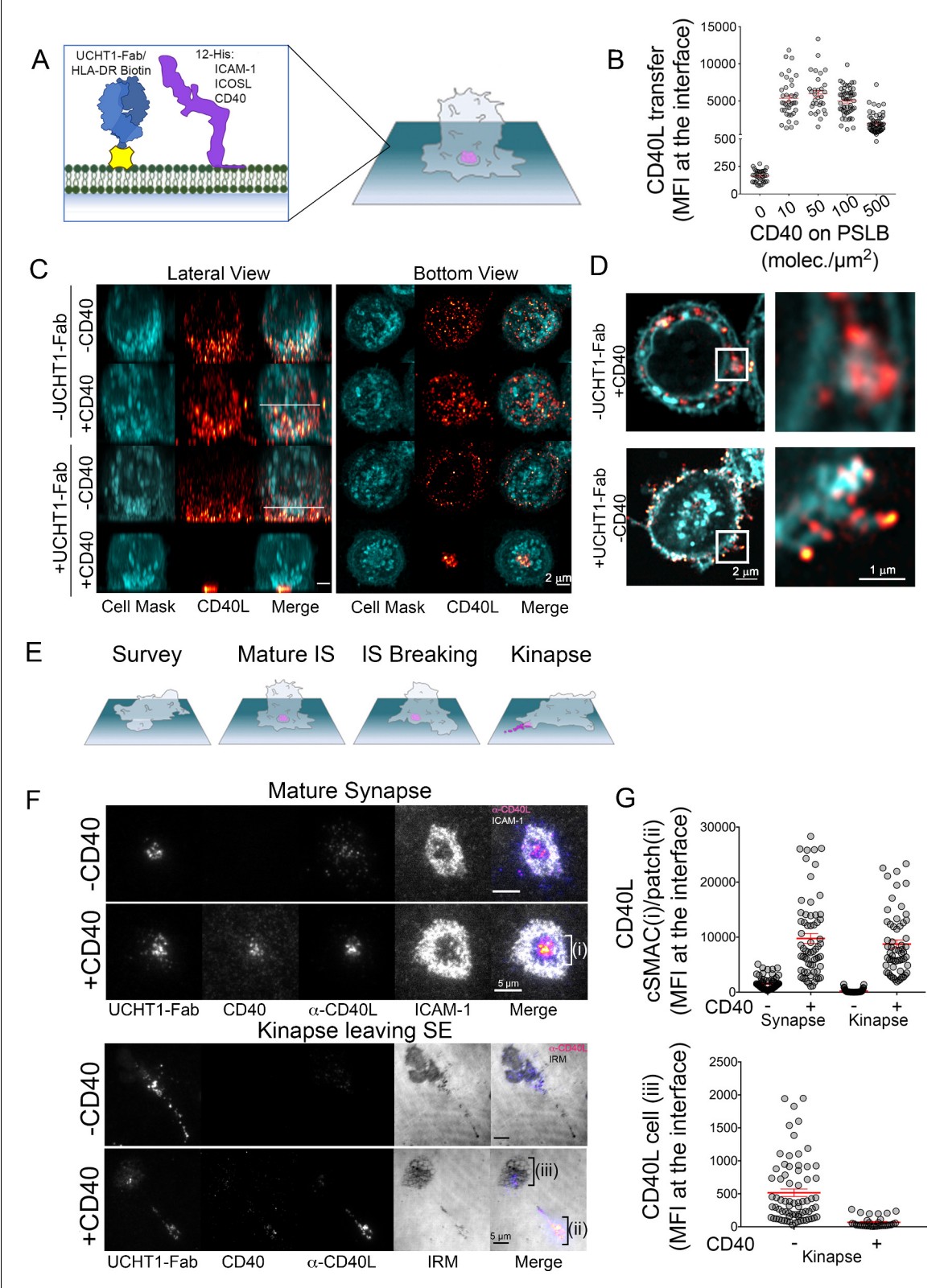

**Figure 1.** CD40 dependent recruitment of CD40L to the IS and deposition in SE trail. (**A**) Schematic of PSLB and mature IS. (**B**) Detection of CD40L with the anti-CD40L clone 24–31 as a function of CD40 in the PSLB. T cells were allowed to form IS for 10 min in the presence of Alexa Fluor 647 anti-CD40L antibody and imaged by TIRFM. Data is pooled from five donors with each point being one cell. (**C**) Representative normalized maximum projections of Airyscan of CD4 [+] T (CellMask, cyan) cell and CD40L (anti-CD40L Alexa Fluor 647, Red hot) on PSLB in the presence/absence of UCHT1-Fab and CD40,

*Figure 1 continued on next page*

*Figure 1 continued*

Scale bar: 2 µm. (**D**) Representative horizontal planes (along the white lines depicted in (**C**)) of CD4 $^+$ T cells showing localization of CD40L within the cell volume. White squares represent the region of interest magnified on the right. (**E**) IS and kinapse stages of T cell interaction. Stages of TCR positive SE are released at the synaptic cleft upon mature IS formation. Following symmetry breaking the SE are partly dragged by the kinapse as they are left (*Choudhuri et al., 2014*). (**F**) Representative TIRFM of IS (top, 10 min incubation) and kinapse (bottom, 90 min incubation) showing CD40 clustering in PSLB coated with ICAM-1, UCHT1-Fab in the presence or absence of CD40. Following fixation and permeabilization cells were stained with anti-CD40L, scale bar: 5 µm. (**G**) Detection of CD40L with anti-CD40L mAb clone 24–31 in (**F**) (****p ≤ 0.0001) nonparametric Mann-Whitney test (U test). Data is from five donors.

DOI: https://doi.org/10.7554/eLife.47528.002

The following figure supplement is available for figure 1:

**Figure supplement 1.** Normalized maximum projections of Airyscan of CD40L (anti-CD40L Alexa Fluor 657, Red hot) within CD4$^+$ T cell volume PSLB in the presence/absence of UCHT1-Fab and CD40, Scale bar: 5 µm.

DOI: https://doi.org/10.7554/eLife.47528.003

of signal at high CD40 density is due to competition between CD40 and the anti-CD40L mAb or some other process, we conclude that CD40L can be detected and localized over the entire physiological range of CD40 densities using anti-CD40L antibody. To investigate the cellular localization of all CD40L, T cells were incubated on the PSLB with ICAM-1 and UCHT1-Fab without or with 50 CD40 molec./µm² for 30 min, fixed, permeabilized and stained with anti-CD40L (Red hot) and Cell-Mask (cyan) to track cell membranes and 3D images generated by super-resolution Airyscan confocal microscopy. On PSLB with ICAM-1 only, most of the CD40L signal was intracellular with rare evidence of CD40L puncta at or near the cell surface based on comparison to the CellMask signal and adding CD40 in the bilayer did not alter this profile (*Figure 1C*, *Figure 1—figure supplement 1*, *Video 1*). On PSLB presenting ICAM1 and UCHT1-Fab, but without CD40, the cell interior was mostly depleted of CD40L and CD40L puncta were distributed over or near the cell surface, often appearing at the ends of small projections (*Figure 1C* and *Figure 1—figure supplement 1*). On PSLB with ICAM-1, UCHT1-Fab and CD40, most of the CD40L was concentrated in the center of the IS and appeared to be just outside the Cell-Mask signal (*Figure 1C* and *Figure 1—figure supplement 1*). On PSLB with ICAM-1 and CD40 but no UCHT1-Fab, CD40L was present in the intracellular compartment (*Figure 1D*, top) whilst CD40L localized to the cell surface and microvilli when PSLB were coated with ICAM-1 and UCHT1-Fab but no CD40 (*Figure 1D* bottom, *Video 2*). Live microscopy demonstrated that

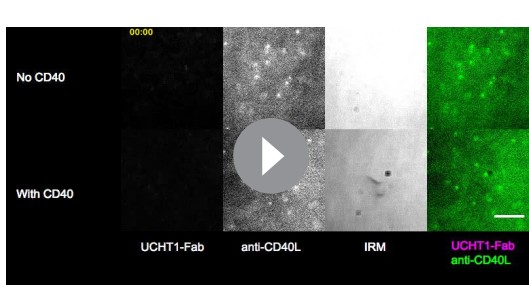

**Video 1.** Live TIRFM imaging of CD40L at the IS. CD4$^+$ T cells were incubated in the presence of anti-CD40L antibody with PSLB coated with ICAM-1, 30 molec./µm2 of UCHT1-Fab in the presence or absence of CD40 at 37C and imaged for the first 15 minutes after contact with the PSLB.

DOI: https://doi.org/10.7554/eLife.47528.004

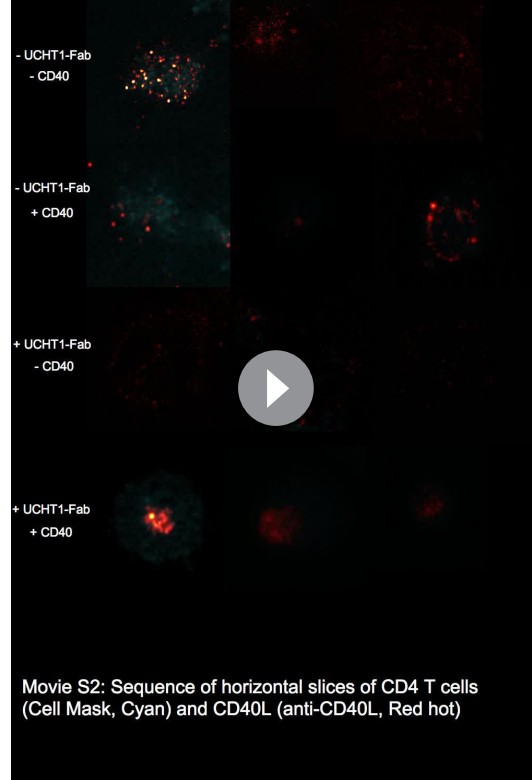

**Video 2.** Sequence of horizontal slices of CD4$^+$ T cells.

DOI: https://doi.org/10.7554/eLife.47528.005

CD40L was detected in puncta in the IS just after and together with TCR-UCHT1-Fab complexes (*Video 1*). When T cells break IS symmetry and transition to migratory kinapses they leave trails of SE behind on the substrate (*Choudhuri et al., 2014*) (*Figure 1E*). To investigate the fate of CD40L, we imaged IS and kinapses by TIRFM to avoid detection of intracellular CD40L. The cSMAC of IS contained partially overlapping signals for UCHT1-Fab and anti-CD40L staining as recently described (*Papa et al., 2017*) (*Figure 1F*; top panel; *Videos 1* and *3*). Upon kinapse formation by T cells, a trail of UCHT1-Fab was present without or with CD40 in the bilayer as expected (*Choudhuri et al., 2014*) (*Figure 1F*; bottom panel, *Video 4*). In the absence of CD40 in the PSLB, the few anti-CD40L reactive puncta that were detected remained associated with the migrating T cell (*Figure 1F*). Kinapses formed in the presence of CD40 resulted in a trail of anti-CD40L reactive puncta that remained tethered to the PSLB (*Figure 1G*). These images and quantification are consistent with CD40L release in SE along with or in parallel with the TCR.

## Selective transfer of CD40L and ICOS into SE

SE are internalized by APCs and thus have been inaccessible to analysis (*Choudhuri et al., 2014*). In addition, the bidirectional transfer of membrane derived molecules between APCs and T cells engaged in the bipartite IS confounds the detection of loss of molecules from the T cell (*Choudhuri et al., 2014*; *He et al., 2007*). We developed a BSLB platform using the same compositions as on PSLB (*Figure 2A*) with similar synaptic accumulation of TCR as viewed by Airyscan confocal microscopy (*Figure 2B* vs *Figure 1B*, and *Video 5*). We incubated T cells and BSLB for 1.5 hr at 37°C and then used ice cold PBS/EDTA to inactive LFA-1-ICAM-1 interactions and gentle shearing forces provided by pipetting to separate the T cells from the BSLB. This enabled the quantitative profiling of gain of T cell molecules on BSLB and loss of molecules from the T cell surface by flow cytometry with a gating strategy based on light scattering and a fluorescent lipid in the BSLB to eliminate cells or cell fragments that happened to overlap with the beads in light scattering (*Figure 2—figure supplement 1*). As with the PSLB system, we needed to assess how CD40 on the BSLB would impact detection of CD40L in the IS using the anti-CD40L antibody. We evaluated this by flow cytometry over a range from 0 to 500 CD40 molec./$\mu m^2$ (*Figure 2C*). We determined the relative IS transfer of CD40L (%) between T cells and BSLB using isotype control-corrected geometric mean fluorescence intensities (GMFI) as follows (GMFI$_{BSLB}$ ÷ (GMFI$_{BSLB}$ + GMFI$_{T\ cells}$)) x 100. In the BSLB system, the decreased detection of CD40L using anti-CD40L mAb was not as prominent such that 500 CD40 molec/$\mu m^2$ was used for most experiments (*Figure 2C*). To corroborate the involvement of the endosomal sorting complex required for transport (ESCRT) in CD40L synaptic transfer to BSLB we targeted tumour susceptibility gene 101 (TSG101) and vacuolar protein sorting 4b (VPS4b) using clustered regularly interspaced short palindromic repeats (CRISPR) gene editing in primary human T cells. We have previously shown that these ESCRT machinery components are required for formation or scission, respectively, of TCR enriched extracellular vesicles (*Choudhuri et al., 2014*). CRISPR/Cas9-gRNA RNP complexes were introduced into T cells by electroporation leading to efficient loss of TSG101 and VPS4b from T cells (*Figure 2D,E* and *Figure 2—figure supplement 1*). As expected, editing out TSG101 and VPS4b led to a significant decrease of CD40L and TCR transfer to BSLB (*Figure 2D and E*), but had no effect on the weak synaptic transfer of CD4 (*Figure 2D and E*). We next tested a larger panel of T cell surface molecules and calculated a relative IS transfer as above to assess selectivity of the sorting process leading to transfer of TCR, CD40L and other surface proteins by human CD4$^+$ T cell blasts to BLSB presenting ICAM-1, UCHT1-Fab with or without combinations of CD40 and ICOSL representing an activated APC (*Figure 2F*, *Figure 2—figure supplement 2*, *Supplementary file 2A*). Transfer of TCR was ligand dose dependent and independent of

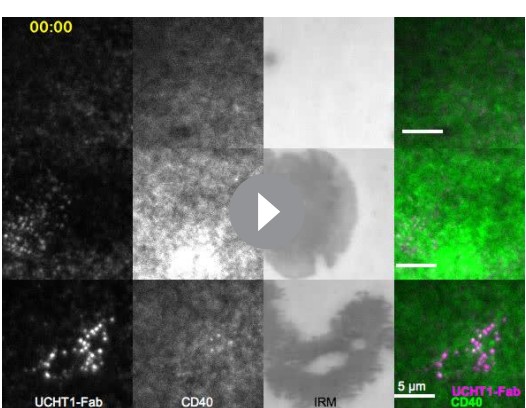

**Video 3.** Live TIRFM of CD4$^+$ T cells showing CD40 clustering at the IS.
DOI: https://doi.org/10.7554/eLife.47528.006

ICOSL and CD40. ICOS was weakly transferred in response to ICOSL without UCHT1-Fab in the BSLB, but the efficiency was increased by TCR engagement. CD40L was transferred only when UCHT1-Fab and CD40 was present in the BSLB, and we were not able to detect a substantial increase when ICOSL was also included in the BSLB in contrast to prior results with PSLB (*Papa et al., 2017*) (*Figure 2—figure supplements 2* and *3*). However, ICOSL presentation on BSLB increased conjugate formation between T cells and BSLB and also enhanced transfer of ICOS and tetraspanins (more significantly CD63) from T cell to BSLB in a manner depending upon the density of ICOSL and the dose of UCHT1-Fab (*Figure 2—figure supplements 2–4*). We detected the enrichment of some additional proteins compared to the plasma membrane from which these vesicles bud. The most enriched (>10% transfer) class of non-ligated proteins are BST-2 (Tetherin) and the tetraspanins CD9, CD63, CD81 and CD82 (*Figure 2D* and *Figure 2—figure supplements 2* and *3*). HLA-ABC was enriched in SE at an intermediate level (1–10%) along with other proteins including CD2, CD6 and CD49D). LFA-1, CD38 and LAMP-1 (CD107a)were represented at < 1% and were not enriched above the ordinary levels found in the plasma membrane (*Figure 2F* and *Figure 2—figure supplement 3D*). To further test the specificity of enrichment we investigated the transfer of other T cell expressed proteins, including OX40, CD27 and GITR. None of these proteins were enriched on BSLBs, at least in the absence of ligands on the BSLB, demonstrating ligand-binding specificity as a dominant requirement for transfer. Its notable that LFA-1 is ligated by ICAM-1 on the BSLB and was not enriched in the SE from the CD4+ effector T cells (*Figure 2F*).

We next determined the location of new proteins detected in the BSLB system in IS formed on PSLB containing UCHT1-Fab, ICAM-1, ICOSL and CD40. BST-2, CD63, CD81, CD82 and CD40L all associated with the cSMAC (*Figure 2—figure supplement 5A*). Tetraspanins CD81, CD63 and CD82 overlapped precisely with UCHT1-Fab with a Pearson Correlation coefficient (PCC) 0.90, 0.86 and 0.90, respectively, whereas BST2 partially overlapped with UCHT1-Fab with a PCC of 0.62 (*Figure 2—figure supplement 5A*). The centers of the punctate UCHT1-Fab and CD40L signals were systematically displaced from each other, but not fully resolved (*Figure 2—figure supplement 5A, B*). Thus, all the SE associated signals from the BSLB transfer experiments localized in the cSMAC in the PSLB system as predicted, but co-localization was variable.

To extend our findings to the context of a physiological ligand for TCR, we investigated the dynamics of SE enrichment using an antigen specific helper T cell clone reactive to HLA-DRB1*09:01-influenza $HA_{338-355}$ (*Figure 2G* and *Figure 3*). We used HLA-DRB1*09:01 loaded with CLIP peptide as a non-agonist pMHC control, as well as UCHT1-Fab as a positive control. CD40L, TCR and BST2 were specifically transferred to BSLB coated with HLA-DRB1*09:01-influenza $HA_{338-355}$ and UCHT1-Fab compared to HLA-DRB1*09:01:CLIP. On PSLB, CD40L localizes to the center of the IS predominantly in the presence of CD40 and HLA-DRB1*09:01-influenza $HA_{338-355}$, not in the presence of HLA-DRB1*09:01:CLIP (*Figure 3A,B*). BST2 was also co-localized in the TCR in an antigen dependent manner (*Figure 3C,D*). As with the polyclonal CD4+ T cells, some ICOS transferred to BSLB with ICOSL with control HLA-DRB1*09:01:CLIP or in the absence of any MHC molecules (*Figure 2G*), a phenomenon which is accompanied by the recruitment of ICOSL to a TCR independent cSMAC-like structure on the PSLB (*Figure 3C,E*). This ICOSL driven TCR independent synapse may exert some control over migration of T cells, but it did not lead to CD40L transfer.

Activated T cells have been shown to transfer CD40L to B cells that expressed CD40, but lacked cognate peptide-MHC in vitro (*Gardell and Parker, 2017*). We thus wanted to ask if activated human T cells were also capable of transferring CD40L to BSLB that lack UCHT1-Fab, but present CD40. We prepared UCHT1-Fab presenting BSLB$^{Atto-488}$ and UCHT1-Fab negative BSLB$^{Atto-565}$ in the four possible combinations where each either presents or does not present CD40 (*Figure 4A*). TCR and CD40L were readily detected on the UCHT1-Fab and CD40 bearing BSLB at 1.5 hr and 24 hr (*Figure 4A*). The surface expression of CD40L on the T cell was detectable at 1.5 hr and 24 hr and was decreased when CD40 was also present on the BSLB with UCHT1-Fab. No CD40L was detected on BSLB when CD40 was not present (*Figure 4A*). This high degree of specificity suggests that CD40L transfer is tightly linked to IS formation, but we wanted to further investigate strongly general activation could trigger CD40L transfer to ICAM-1 and CD40 bearing SLB in the absence of TCR engagement. We followed two approaches. First, we incubated T cells with phorbol myristate acetate (PMA) and ionomycin for 30 min to expose CD40L on the surface and then for another 90 min in the presence of BSLBs with ICAM-1 and CD40 only or ICAM-1, UCHT1-Fab and CD40. PMA-ionomycin significantly increased the relative transfer of CD40L to BSLB with 0 or 20 molec./$\mu m^2$ UCHT1-

**Video 4.** Live TIRFM of CD4 [+] Tcells showing kinapse formation on PSLB.
DOI: https://doi.org/10.7554/eLife.47528.007

Fab (*Figure 4B*). This demonstrates that TCR engagement is not absolutely necessary for CD40L transfer.

## Mass Spectrometry (MS) of SEs reveals enrichment of ESCRT proteins and TCR signaling

TCR-enriched SE are released through a TSG101 and VPS4-dependent plasma membrane budding process (*Choudhuri et al., 2014*). Both TSG101 and VPS4 form part of the Endosomal Sorting Complex Required for transport (ESCRT). Specifically, TSG101 (an ESCRT-I member) is required for TCR sorting into membrane buds (*Vardhana et al., 2010*), whilst VPS4 mediates scission from the plasma membrane and release into the cSMAC/synaptic cleft (*Choudhuri et al., 2014*). Residual ESCRT components may be trapped in the SE and the repertoire of ESCRT components involved in this process expanded by MS analysis of a sorted BSLB to enrich SE. We allowed 50 million CD4[+] effector T cells to form IS on BSLB with ICAM-1, CD40, ICOSL in the presence or absence of UCHT1-Fab, disengaged the T cells and BSLB by incubation with ice-cold PBS/EDTA, sorted 5 million BSLB and subjected them to MS analysis after extraction of lipids. STRING analysis of 130 candidates having an absolute fold change (+UCHT1-Fab /- UCHT1 Fab) of at least 3.35 revealed a network of proteins with reported interactions. They are displayed to minimize the energy of the system based on the confidence score for interactions (*Figure 5A*, supplement excel file 1). A Markov clustering algorithm (*Enright et al., 2002*) highlighted six modules with three or more nodes displaying a high degree of adjacency (*Figure 5B*). We used Reactome (reactome.org) to determine if more proteins are found for particular pathways than is predicted by chance. This analysis revealed significant enrichment of TCR signaling, vesicle mediated transport and ESCRT machinery pathways (*Figure 5C*). Several surface proteins identified as enriched in the immunofluorescence analysis (*Figure 2D*) were also enriched in SE as defined by MS including CD40L, ICOS, CD81, CD82, BST2, HLA-A, HLA-B, CD2, CD6 and CD49d (ITGA4/ITGB1) (*Figure 5A*, and supplement excel file 1). Most of the surface proteins that were not enriched in the immunofluorescence analysis (*Figure 2D*) were also not detected in the MS analysis including CD4, CD27, CD38, CD45, CD80, OX40, GITR and LAMP1 (*Figure 5A*, and supplement excel file 1). However, a few proteins not enriched in SE by immunofluorescence analysis (*Figure 2D*) including LFA-1 (ITGAL/ITGB2) and CD28 were enriched in SE by MS (*Figure 5A*, and supplement excel file 1). However, reexamination of the raw immunofluorescence data reveals that both LFA-1 and CD28 gave a stronger signal on BSLB with UCHT1-Fab compared to without UCHT1-Fab, even though only 0.2% of the surface protein was transferred in the presence of UCHT1. Thus, the MS analysis fully agreed with the immunofluorescence analysis. ESCRT-0 components HRS, STAM2 and EPN1, ESCRT-II component ALIX ESCRT-III component CHMP4b were enhanced on BSLB by TCR engagement (*Figure 5A*, and supplement excel file 1). Within the range of TIRF microscopy, EPN1 and ALIX were localized in or around the cSMAC (*Figure 5D,E*), whereas STAM2 was present in the cSMAC, but also in peripheral components of the IS, consistent with additional functions of STAM2 in the IS (*Bache et al., 2003*). Airyscan confocal microscopy revealed that EPN1 was co-localized with UCHT1-Fab in the cSMAC formed on PSLB also presenting ICAM-1 together with CD40 (*Figure 5—figure supplement 1*). We next investigated the effect of suppressing ALIX, EPN1 and CHMP4b by direct delivery of CRISPR/CAS9-gRNA RNP complexes in CD4[+] T cell blasts prior to assessing transfer of CD40L, TCR and CD81 to BSLB presenting ICAM-1, CD40 and incremental levels of UCHT1-Fab (*Figure 5F* and *Figure 5—figure supplement 2*). Both EPN1 and CHMP4b suppression resulted in a significant reduction of CD40L transfer to BSLB, which was not observed for TCR (*Figure 5F* and *Figure 5—figure supplement 2*). Interestingly, a similar trend was observed for Alix, which slightly reduced CD40L but not TCR transfer, suggesting a differential role of EPN1, CHMP4b and ALIX in the regulation of sorting and release of CD40L on SE (*Figure 5F* and *Figure 5—figure supplement 2*). As expected, the synaptic transfer of CD81 was also reduced by suppression of ALIX, EPN1, CHMP4b, TSG101 and VPS4b proteins (*Figure 5—figure supplement 2*). On the other hand, suppression of ALIX had no effect on transfer of TCR and CD40L, suggesting a potential degree of redundancy of ESCRT involved in the release of SE at the IS.

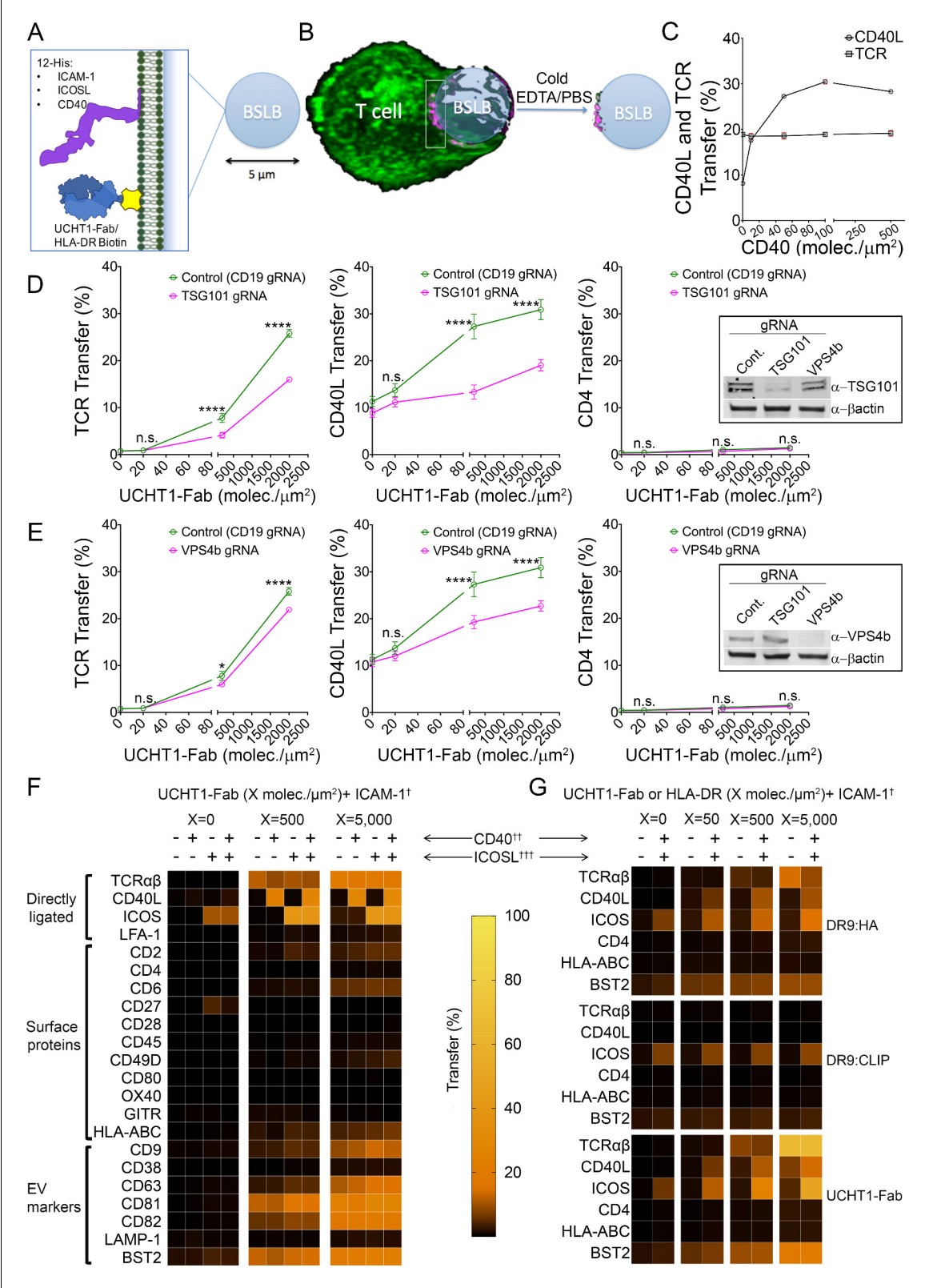

**Figure 2.** CD40L is incorporated in SE. (**A**) Schematic of BSLB (**B**) Schematic of T cell (green) interacting with BSLB with UCHT1-Fab (magenta) and SE on BSLB after T cell removal. (**C**) % CD40L and TCR transfer to BSLB coated with incremental levels of CD40. Data is from six donors. (**D**) Percentage transfer of TCR, CD40L and CD4 to BSLB following electroporation of CD4⁺ T cells with either control CRISPR/Cas9-CD19gRNA RNP or CRISPR/Cas9-TSG101gRNA RNP, insert WB showing degree of knockdown using anti-TSG101 antibody. (**E**) Percentage transfer of TCR, CD40L and CD4 to BSLB

*Figure 2 continued on next page*

*Figure 2 continued*

following electroporation of CD4+ T cells with either control CRISPR/Cas9-CD19gRNA RNP or CRISPR/Cas9-VPS4bgRNA RNP, insert WB showing degree of knockdown using anti-VPS4b antibody. Multiple t-test *P* values < 0.05 (*);<0.0001 (****) were considered significant; n.s. = non significant. (**F**) Heat maps showing the percentage of proteins transferred from T cells to BSLB. Data is from 10 donors. (**G**) Heat maps showing the percentage of proteins transferred from clone 35 to BSLB. Data is representative of 3 independent experiments with different clone 35 aliquots.

DOI: https://doi.org/10.7554/eLife.47528.008

The following figure supplements are available for figure 2:

**Figure supplement 1.** Gating Strategy for SE on BSLB and representative histograms following CRISPR/Cas9 knockdown.

DOI: https://doi.org/10.7554/eLife.47528.009

**Figure supplement 2.** ICOS significantly increases CD40L transfer to BSLB.

DOI: https://doi.org/10.7554/eLife.47528.010

**Figure supplement 3.** Efficient transfer of CD40L to BSLB at low UCHT1-Fab densities.

DOI: https://doi.org/10.7554/eLife.47528.011

**Figure supplement 4.** ICOSL increases T cell: BSLB conjugate formation and ICOS transfer.

DOI: https://doi.org/10.7554/eLife.47528.012

**Figure supplement 5.** BST2, CD63, CD81 and CD82 and CD40L localize to the synaptic cleft of UCHT-1 Fab stimulated cells.

DOI: https://doi.org/10.7554/eLife.47528.013

MS also revealed enrichment of disintegrin and metalloproteinase domain-containing protein 10 (ADAM10), which mediates CD40L release as a soluble trimer from T cells upon CD40 engagement (*Yacoub et al., 2013*). We therefore evaluated the transfer of CD40L together with TCR and CD81 in the presence of metalloproteinase inhibitors TAPI-2 and GI254023X as well as through direct suppression of ADAM10 via delivery of CRISPR/CAS9-ADAM10 gRNA RNP complex (*Figure 5—figure supplement 3*). GI254023X which displays a higher selectivity for ADAM10 but not TAPI-2 resulted in more CD40L transfer to BSLB. A similar increase in CD40L transfer to BSLB was observed following suppression with CRISPR/CAS9-ADAM10 gRNA RNP complex. This result suggests that metalloprotease dependent cleavage of CD40L is not required for synaptic transfer of vesicular CD40L to BSLB. The BSLB system thus expands our understanding of SE composition, including candidates for ESCRT pathway function and other aspects of SE biology.

## TCR and CD40L occupy spatially distinct microclusters within single SE

We next investigated the nanoscale organization of protein microclusters in SE deposited on PSLB after gentle removal of helper T cells (*Figure 6A*). We applied three-color dSTORM to localize SE proteins with a resolution of 20 nm. To visualize SE we took advantage of the fact that these membranous structures are enriched in glycoproteins compared to the sparsely populated PSLB and used wheat germ agglutinin (WGA) labeling to provide contrast. In agreement with earlier electron microscopy measurements (*Choudhuri et al., 2014*), the average diameter of SE was 84 ± 5 nm (*Figure 6B–D*) and 36 ± 3 SE were transferred per IS (*Figure 6E*). The number and size of TCR microclusters transferred on SEs is similar to that of TCR microclusters present in the early IS (*Figure 6F–H*). This provides further evidence that TCR microclusters are converted into SE. To determine the localization of protein microclusters the SE were stained with anti-CD81-CF568 (*Figure 7A*), anti-TCRαβ-AF488 and a mAb for BST2, CD40L, or ICOS conjugated with AF647. Three color dSTORM revealed 3 subsets of SE: 1) TCR only, 2) CD40L, ICOS or BST2 only and 3) double positive for TCR and CD40L (54.5%), ICOS (77.5%) or BST2 (75.0%) (*Figure 7B* and *Figure 7—figure supplements 1–3*). To address the degree of colocalization we applied two independent methods: coordinate-based colocalization (CBC) (*Malkusch et al., 2012*) and cross-correlation (*Stone et al., 2017*). TCR clusters were colocalized with ICOS or BST2 within a search radius of 50 nm, and segregated from CD40L clusters

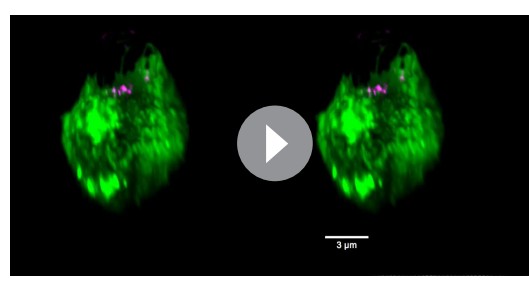

**Video 5.** Airyscan of CD4+ T cell (green) interacting with BSLB coated with UCHT1-Fab (magenta) and ICAM-1.

DOI: https://doi.org/10.7554/eLife.47528.014

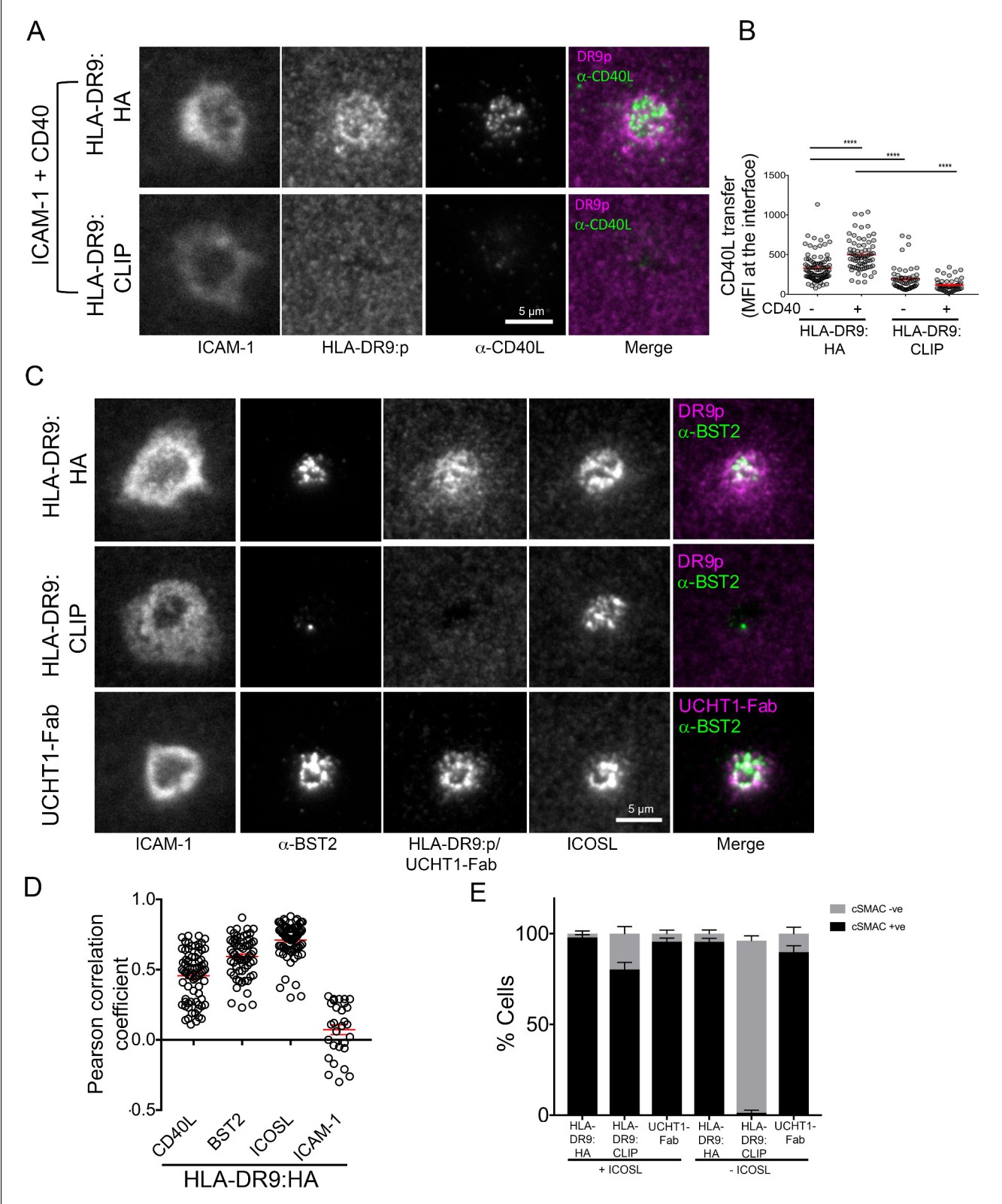

**Figure 3.** CD40L, BST2, and ICOSL localize to the synaptic cleft. (**A**) Representative TIRFM images and IRM images showing staining of CD40L in the IS following incubation of HA specific clones on PSLB coated with ICAM-1 and either HLA-DR9$_{HA}$ or HLA-DR9$_{CLIP}$ monomers. Scale bar: 5 μm. (**B**) Detection of CD40L staining from (**A**) expressed as arbitrary unit (A.U.). ($^{****}$p ≤ 0.0001) nonparametric Mann-Whitney test (U test). Data is from three experiments with different clone 35 aliquots. (**C**) Representative TIRFM and IRM images showing staining of BST2 in the IS following incubation of HA

*Figure 3 continued on next page*

Figure 3 continued
specific clones on PSLB coated with ICAM-1 and either His-tagged UCHT1-Fab, HLA-DR9$_{HA}$ or HLA-DR9$_{CLIP}$ monomers. Scale bar: 5 µm. (D) Pearson correlation of CD40L, BST2, ICOSL and ICAM-1 with HLA-DR9$_{HA}$ (as seen in C). (E) Percentage cells with or without cSMAC like structure (as seen in C) following incubation of HA specific clones on PSLB coated with ICAM-1 and either His-tagged UCHT1-Fab, HLA-DR9$_{HA}$ or HLA-DR9$_{CLIP}$ monomers in the presence or absence of ICOSL.

DOI: https://doi.org/10.7554/eLife.47528.015

(*Figure 7C*). There was a high correlation between TCR with ICOS or BST2 at inter-particle distances less than 50 nm, whereas TCR and CD40L showed the highest correlation at a distance of 150 nm (*Figure 7D* and *Figure 7—figure supplement 3*). Corroborating this, the mean nearest-neighbor distance (NND) of paired single-molecule localizations was 34 nm for TCR and BST2, 17 nm for TCR and ICOS and 230 nm for TCR and CD40L (*Figure 7E*). As a positive control, TCR molecules on SE detected with anti-TCRαβ−AF488 and UCHT1-Fab-AF647 showed a high degree of colocalization (*Figure 7C–E*). The results are consistent with TCR, BST2 and ICOS occupying overlapping microclusters, whereas TCR and CD40L occupied spatially distinct microclusters that are sorted within

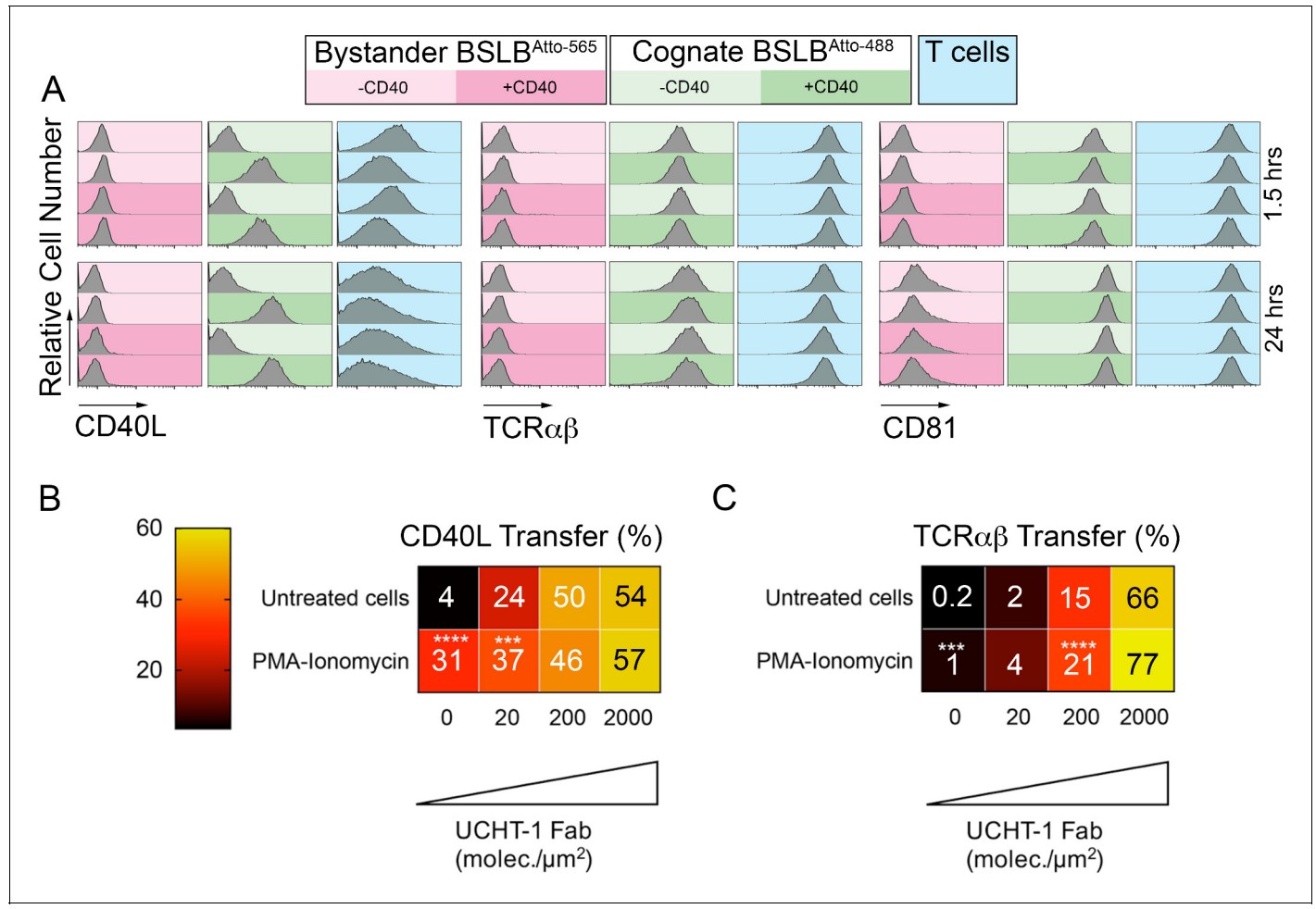

**Figure 4.** Specificity of CD40L transfer- effects of bystander BSLB and general activation. (A) Representative flow cytometry histograms and percentage marker transfer (CD40L, TCR and CD81) of 'cognate' (UCHT1-Fab +ve Atto488; green) and bystander (UCHT1-Fab –ve Atto565; red) BSLB and T cells (blue). (B,C) Multiple t-test to compare the relative synaptic transfer (%) of CD40L (B) and TCRαβ (C) between cells pulsed for 30 min with PMA-Ionomycin (10 ng/mL and 0.5 µg/mL) and then incubated for another 90 min with agonistic BSLBs (increasing densities of UCHT-1-Fab, CD40 20 molec./µm$^2$, ICAM-1 200 molec./µm$^2$) in the presence of the PMA-Ionomycin (values inside each cell represent mean percent synaptic transfer of 6 donors). Untreated cells were used as controls and as reference group for statistics; *, p≤0.05; ***p≤0.001; ***p≤0.0001 (data is from six donors).
DOI: https://doi.org/10.7554/eLife.47528.016

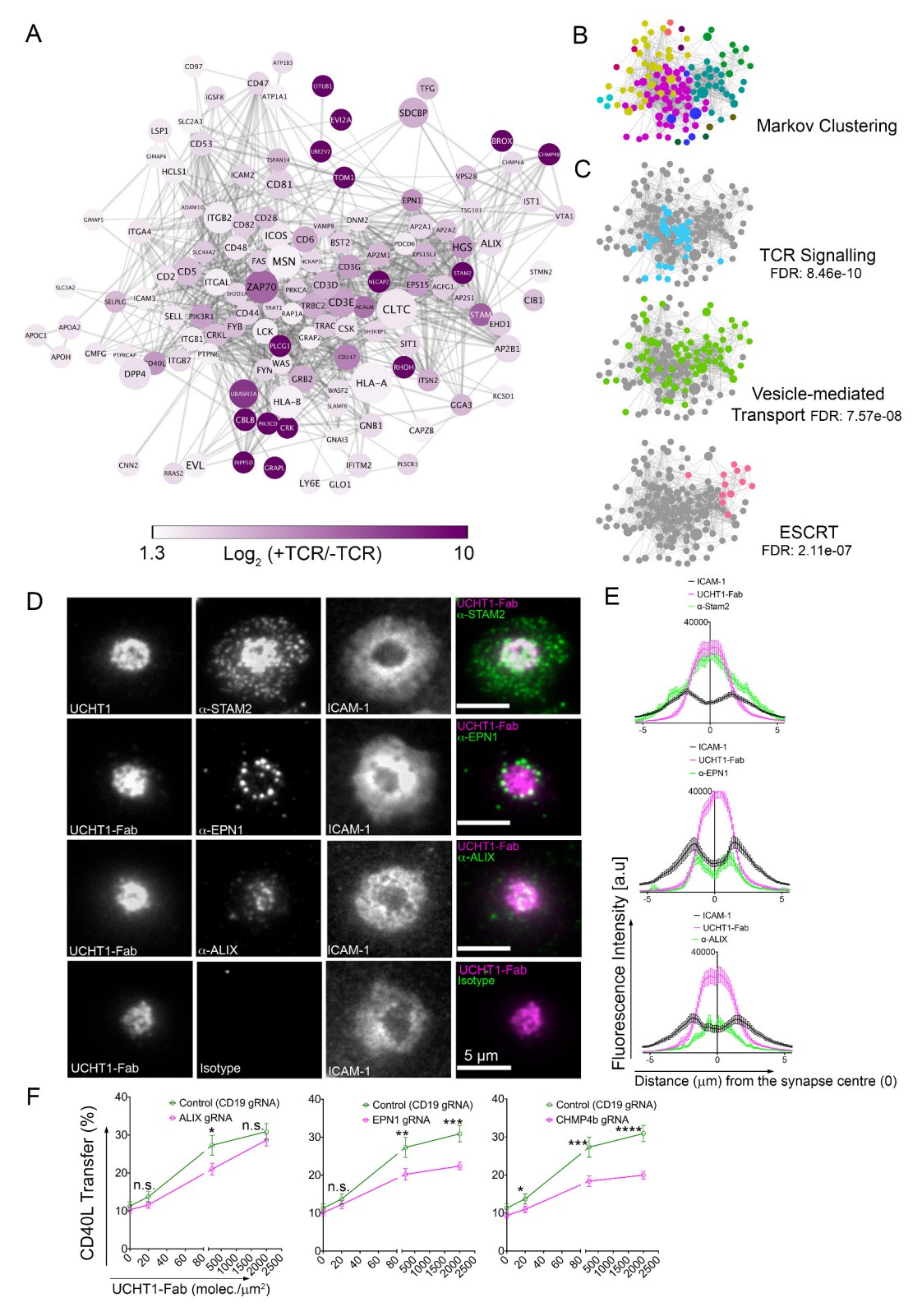

**Figure 5.** SE contains ESCRTs and TCR signalosome. (**A**) Proteins enriched by UCHT1-Fab on BSLB also containing ICAM-1, ICOSL and CD40. The network plot is based on known and predicted interactions from the STRING database (v11), with minimal confidence score of 0.4. Each protein displayed > 1.75 log2 fold enrichment over BSLB coated with ICAM-1, ICOSL and CD40 in two independent experiments (three pooled donors/experiment). Node area represents protein LFQ intensity. Line thickness represents confidence score (0.4–0.999). False discovery rate (FDR) on peptide

*Figure 5 continued on next page*

*Figure 5 continued*

and protein level were set to 1%. (B) Modules identified in protein network shown in (A) determined using the Markov Cluster algorithm (inflation parameter: 2.5). Each colour represents a separate module (associated adjacency matrix). (C) Reactome analysis (reactome.org) of protein network shown in (A) reveals enrichment of TCR signaling, Vesicle-mediated transport and ESCRT pathways. (D) Representative TIRFM and IRM images showing staining of STAM2, EPN1 and ALIX on PSLB with UCHT1-Fab and ICAM-1, CD40 and ICOSL. T cells were incubated with PSLB at 37°C for 1 hr. Following fixation and permeabilization, cells were stained with relevant antibodies. (E) Average radial distribution in IS formed on PSLB containing ICAM-1 (black), ICOSL, CD40 and UCHT1-Fab (magenta) stained with antibodies to the candidate proteins (green). Data are from 3 donors and 50 cells; scale bar: 5 μm. (F) Percentage transfer of CD40L to BSLB following electroporation of CD4$^+$ T cells with either control CRISPR/Cas9-CD19gRNA RNP or CRISPR/Cas9-VPS4bgRNA RNP. Multiple t-test, *P* values < 0.05 (*);<0.002 (**);<0.0002 (***);<0.0001 (****) were considered significant; n. s. = non significant.

DOI: https://doi.org/10.7554/eLife.47528.017

The following figure supplements are available for figure 5:

**Figure supplement 1.** Recruitment of EPN1 to the plasma membrane.
DOI: https://doi.org/10.7554/eLife.47528.018
**Figure supplement 2.** Transfer to BSLB following CRISPR/Cas9 RNP electroporation.
DOI: https://doi.org/10.7554/eLife.47528.019
**Figure supplement 3.** Effect of ADAM10 inhibitor and CRISPR/Cas9 electroporation on transfer to BSLB.
DOI: https://doi.org/10.7554/eLife.47528.020

single SE. Segregation of microclusters within SE and populations of single positive vesicles explain the failure of TCR and CD40L immunofluorescence to co-localize in IS images (*Figure 2—figure supplement 5* and *Figure 3*), although they were otherwise closely linked in the cSMAC.

## Released SE induce DC maturation and cytokine production

We next evaluated whether T cell isolated SE have an activating effect on monocyte-derived dendritic cells (moDCs). We therefore, first generated T cell derived SE on PSLBs coated with a combination of different T cell accessory ligands. After removal of T cells, immature moDCs were incubated with the released SEs for 24 hr followed by analysis of DC maturation markers and secreted factors. Conditions A1 and C0 had no SE, B1 had CD40L low SE, and C1 had CD40L high SE. CD40L high SE triggered high expression levels of HLA-DR and CD83 (*Figure 8A,B* and *Figure 8—figure supplement 1A,B*). Analysis of 105 secreted factors from DC, revealed that CD40L high SE significantly increased release of 46 factors, prominently including several chemokines and inflammatory cytokines such as tumor necrosis factor (TNF) and IL-12p70, which are important pro-inflammatory cytokines from licensed DCs (*Cella et al., 1996*) (*Figure 8C* and *Figure 8—figure supplement 1C*). These results demonstrate that SE have an active form of CD40L that is sufficient to induce DC maturation in the absence of T cells. Another important aspect of SE mediated transfer is that many receptor systems involved in cell-cell communication requires a high order clustering, and therefore valency, to achieve full agonist function (*Haswell et al., 2001*). The higher density of proteins in SE may therefore provide mechanistic insight in which SE promote extensive CD40 cross-linking on DC surface to robustly trigger down-stream signaling. Since each vesicle has an average diameter of 84 nm and there is an average of 37 vesicles released per IS (*Figure 7C and D*), the released surface area of all SE can be estimated as 0.82 μm$^2$. This is ~ 0.2% of the surface area of the T cell, consistent with the % transfer of CD4 and CD38. Given this small surface area, the density of CD40L on SE is estimated to be over 200 trimers/μm$^2$ (*Supplementary file 2C*). As we have noted in the dSTORM imaging, these free CD40L trimers are further organized in even higher density microclusters. This provides a mechanism for SE to promote extensive CD40 cross-linking on the APCs surface to robustly activate down-stream signals. To test this hypothesis, we developed synthetic unilamellar vesicles (SUVs) loaded with increasing quantities of N-terminal His-tagged hCD40L (>200 molec./μm$^2$, *Figure 8D*, *Supplementary file 2C,D*) to corroborate whether high binding valency due to CD40L concentration on sub-100 nm vesicles has a superior agonist effect on APCs. We then compared moDC activation when presented with either i) SUVs loaded with CD40L (vesicular CD40L), ii) soluble CD40L trimers (sCD40L) or iii) soluble CD40L trimers in the presence of unloaded (mock) SUVs. We observed a significant upregulation of HLA-DR, CD40, ICAM-1, CD80 and CD86 on moDC treated with vesicular CD40L compared to an equivalent concentration of

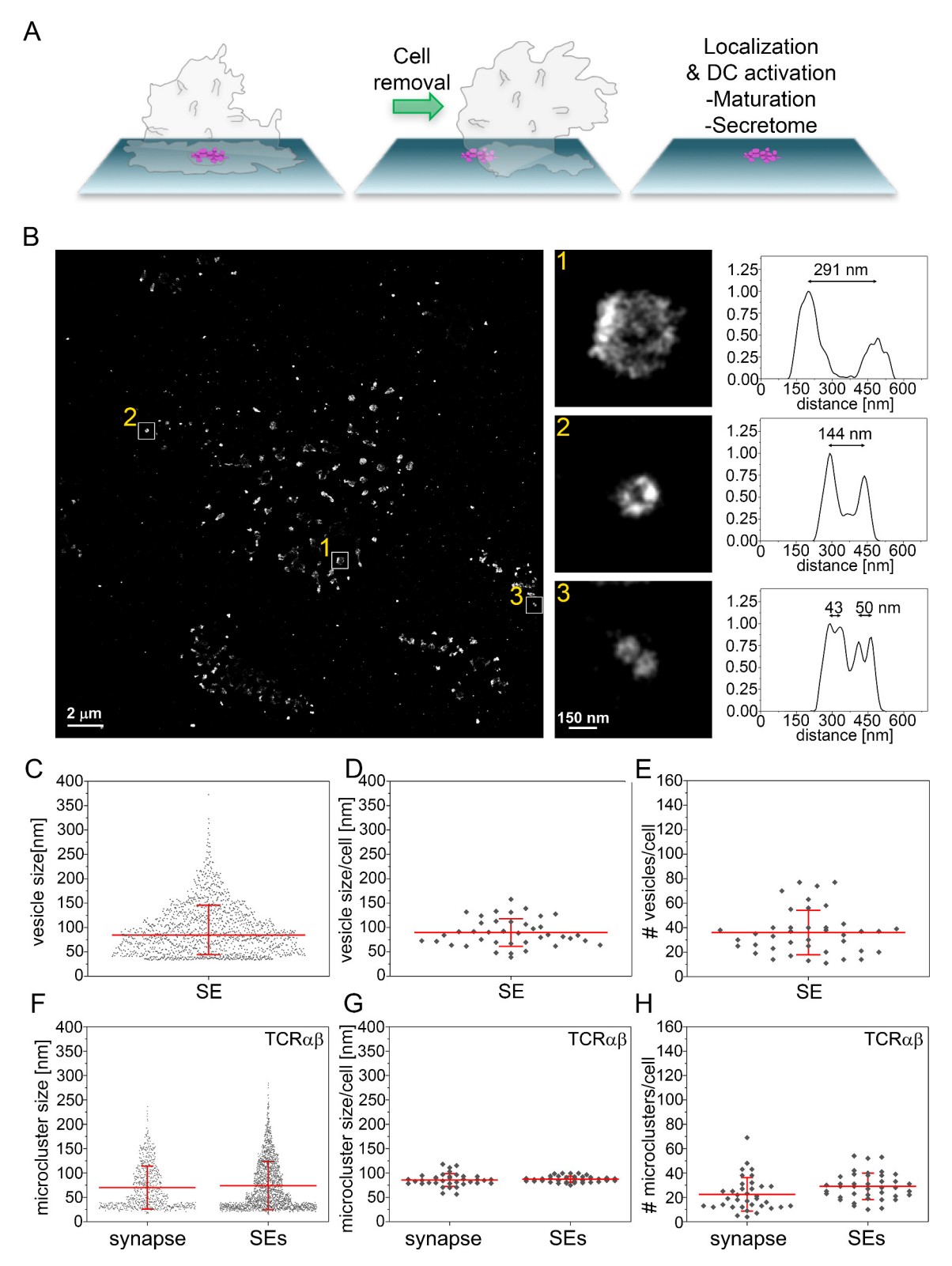

**Figure 6.** Size distribution SE captured from immunological synapses on PSLB. (**A**) Schematic of SE deposition on PSLB. CD4[+] T cell blasts were incubated for 90 min on supported lipid bilayers coated with ICAM-1 (200 molec./μm²), UCHT1-Fab (30 molec./μm²), CD40 (500 molec./μm²) and ICOSL (200 molec./μm²), then T cells were removed with ice cold PBS and fixed with 4% PFA. (**B**) Representative dSTORM image of SE released by T cells were stained with WGA-CF568 to visualize the glycans (carried by lipids and proteins) on the surface of the SE. Examples of SE of different sizes,

*Figure 6 continued on next page*

*Figure 6 continued*
depicted by the white squares, are zoomed-in and shown with correspondent relative fluorescence intensity profiles. (C) Size distribution of SE released from all cells. Each symbol represents an SE (n = 1482). (D) Size distribution of SE released per cell. Each symbol represents the median size of SE released per cell (n = 40). (E) Number of SE released per cell. Each symbol represents the median number of SE released per cell. (F) Size distribution of TCRαβ microclusters from synapse and SE. Each symbol represents a microcluster. (G) Size distribution of TCRαβ microclusters from synapse and SE released per cell. Each symbol represents the mean size of SE released per cell. (H) Number of TCRαβ microclusters from synapse and SE released per cell. Each symbol represents the mean number of microclusters per cell. Lines and errors represent means ± SD.
DOI: https://doi.org/10.7554/eLife.47528.021

soluble CD40L trimers (*Figure 8E–F*) demonstrating that vesicular CD40L is a superior agonist compared to soluble CD40L.

## Discussion

In this study, we systematically profiled composition, mechanism of formation, nanoscale structure and function of SE that were produced and captured in a model IS. Importantly, we adopted BSLB as a scalable platform to analyze SE by flow cytometry and mass spectroscopy. BSLB have been used previously to demonstrate activity of purified MHC proteins (*Gay et al., 1986*), detect weak protein interactions (*Baksh et al., 2004*) and determine biophysical requirements for phagocytosis (*Bakalar et al., 2018*). We have also used BSLB to calibrate site densities of recombinant proteins attached to SLB by flow cytometry (*Dustin et al., 2007*) and for bulk functional assays (*Vardhana et al., 2010*). Here, we allowed T cells to form IS with BSLB at a 1:1 ratio and then disrupted the conjugates to allow analysis of material transferred from the T cell to the BSLB. The strength of this approach is that each BSLB carries the output from one T cell, on average, such that the output of single IS can be quantified and calibrated to number of antibody binding sites and compared to average levels of the same molecules on the donor T cells. Another strength is that BSLB can then be isolated by particle sorting, which enabled MS analyses with minimal cellular contamination. A limitation of the approach is that while we can readily analyze both the cells and BSLB at a population level, we cannot link the cellular donor to a BSLB recipient specifically, which would require a different approach- perhaps with microfluidics. For these studies we used ~ 5 µm beads, which are near cell size and are easily analyzed on commercial flow cytometers. The results we obtained are qualitatively comparable to results with PSLB, but we cannot rule out some impact of the surface curvature of the BSLB. Generally, transfer of molecules to BSLB predicted localization in the cSMAC formed on PSLB. We also had some concern that the methods used to disrupt conjugates (divalent cation chelation and low temperature) might result in physical tearing off of membranes that might not normally be transferred to APC, and would contaminate SE. However, there is a high degree of enrichment for ligated cargo like ICOS and CD40L (>10%), tetraspannins (*Théry et al., 2001*) and BST2 (34) (~10%), and proteins with known tetraspannin association, like MHC class I (1–10%). In contrast, most other proteins are transferred at a level similar to the area of membrane transferred (~0.2%). This suggests a high degree of selectivity of the transfer process and minimal contamination from random pieces of plasma membrane. Furthermore, CRISPR gene editing of TSG101 and VPS4 clearly reduced the transfer of TCR and CD40L, further demonstrating specificity for ESCRT dependent vesicle formation. We expect that BSLB will be a useful approach to study synaptic transfer in other biological systems.

Experiments on PSLB offer further insights into the mechanism of SE formation and cargo selection. We previously had failed to detect tetraspanin CD63 enrichment in SE (*Choudhuri et al., 2014*), but find here tetraspanins CD81 is the most strongly enriched tetraspanin in transfer to BSLB and cSMAC localization on PSLB. CD81 staining also allowed visualization of the complete SE membrane by dSTORM. BST2 may tether some SE to the T cell membrane, accounting for dragging of SE behind kinapses, and may provide a mechanism for NF-κB signaling in the T cell in response to SE formation (*Edgar et al., 2016*; *Neil et al., 2008*). We observed formation of TCR engagement independent IS-like structures and SE-like vesicle transfer stimulated by ICOSL, but these did not elicit CD40L transfer even when CD40 was also present in the BSLB. TCR engagement independent IS are stimulated in CD8 effector T cells by NKG2D ligands, but similarly don't stimulate effector

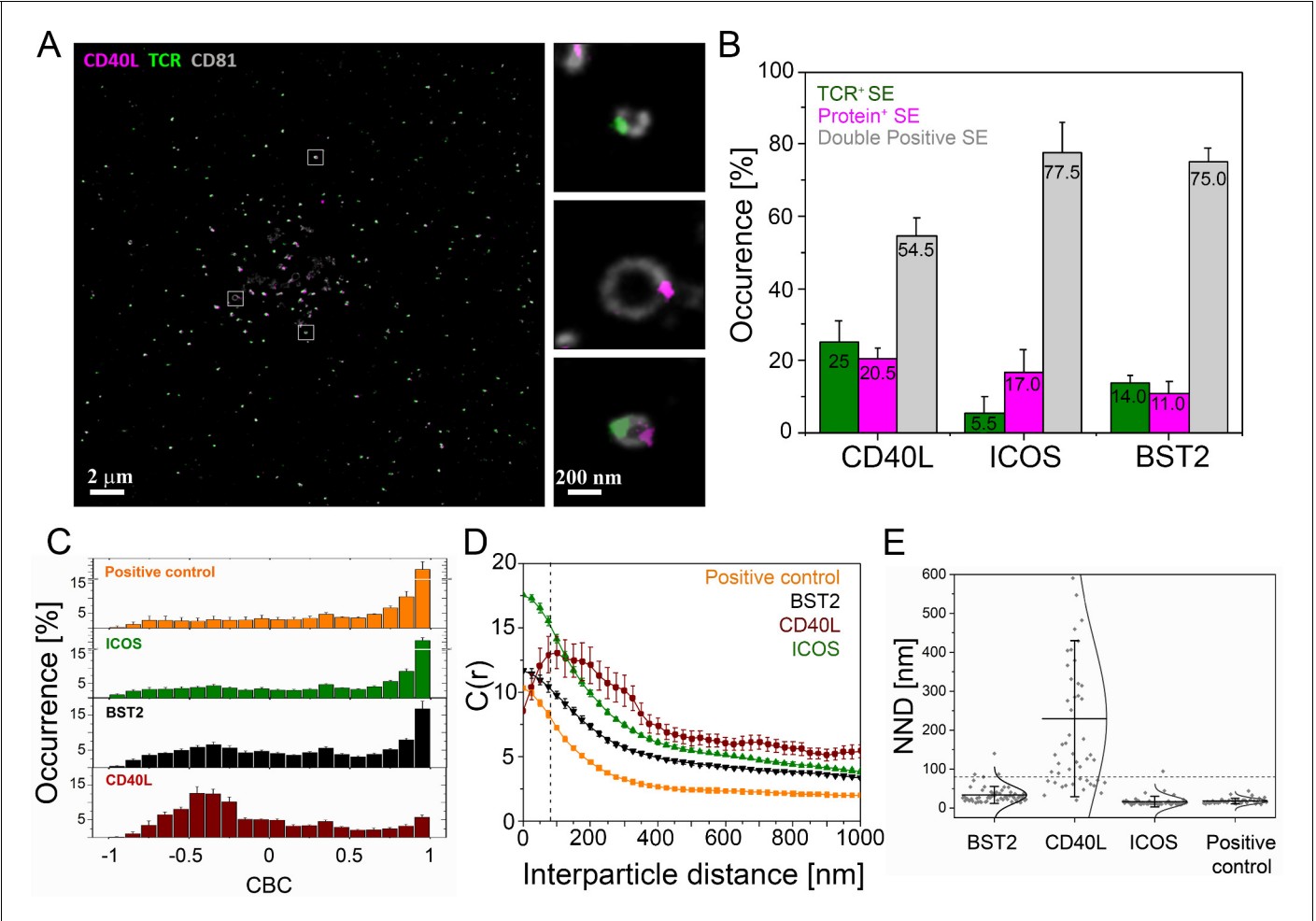

**Figure 7.** Nanoscale structure of SE. CD4[+] T cell blasts were allowed to form IS for 90 min on PSLB with ICAM-1 (200 molec./μm[2]), ICSOL (200 molec./μm[2]), CD40 (500 molec./μm[2]) and UCHT1-Fab (30 molec./μm[2]) and then released with cold PBS and the SE were fixed, stained with mAb as indicated, and subjected to dSTORM analysis. (A) Representative dSTORM images showing TCR (green), CD40L (magenta) on CD81 (gray) labeled SE. Insets show examples of SEs containing only TCR, only CD40L or both proteins. (B) Percentage of SEs containing only TCR (green), only protein of interest (magenta), or containing both TCR and protein of interest (gray). (C) CBC histograms of the single-molecule distributions of the colocalization parameter. Bars represent means ± SD. The positive control is TCRαβ−AF488 and UCHT1-Fab-AF647, which are predicted to co-localize. (D) Cross-correlation analysis. (E) Nearest-neighbor distance (NND) analysis from data shown in B). Each symbol represents the median NND of all paired single-molecule localizations from EVs released per cell. Lines and errors represent means ± SD. Dashed line in panel (D) and (E) marks mean size of SE.

DOI: https://doi.org/10.7554/eLife.47528.022

The following figure supplements are available for figure 7:

**Figure supplement 1.** TCR co-localization with BST2 and ICOS and segregation from CD40L.

DOI: https://doi.org/10.7554/eLife.47528.023

**Figure supplement 2.** CD81 co-localization with TCR and CD40L.

DOI: https://doi.org/10.7554/eLife.47528.024

**Figure supplement 3.** Distinct cross-correlation distances for TCR with BST2 and ICOS versus CD40L.

DOI: https://doi.org/10.7554/eLife.47528.025

function (*Markiewicz et al., 2005*; *Somersalo et al., 2004*). TCR engagement independent IS-like structures may allow T cells to more closely inspect APC for presence of relevant pMHC.

We have demonstrated that SE incorporate CD40L as long as CD40 is included in the SLB in addition to a TCR agonist and ICAM-1. CD40L is primarily expressed on CD4[+] T cells, but low-level mRNA and/or protein expression has also been reported on B cells, basophils, eosonophils, NK cells, macrophages, dendritic cells, smooth muscle cells (*Michel et al., 2017*; *Schönbeck et al., 2000*), and platelets (*Sprague et al., 2008*). Extracellular vesicles from platelets possess CD40L and these

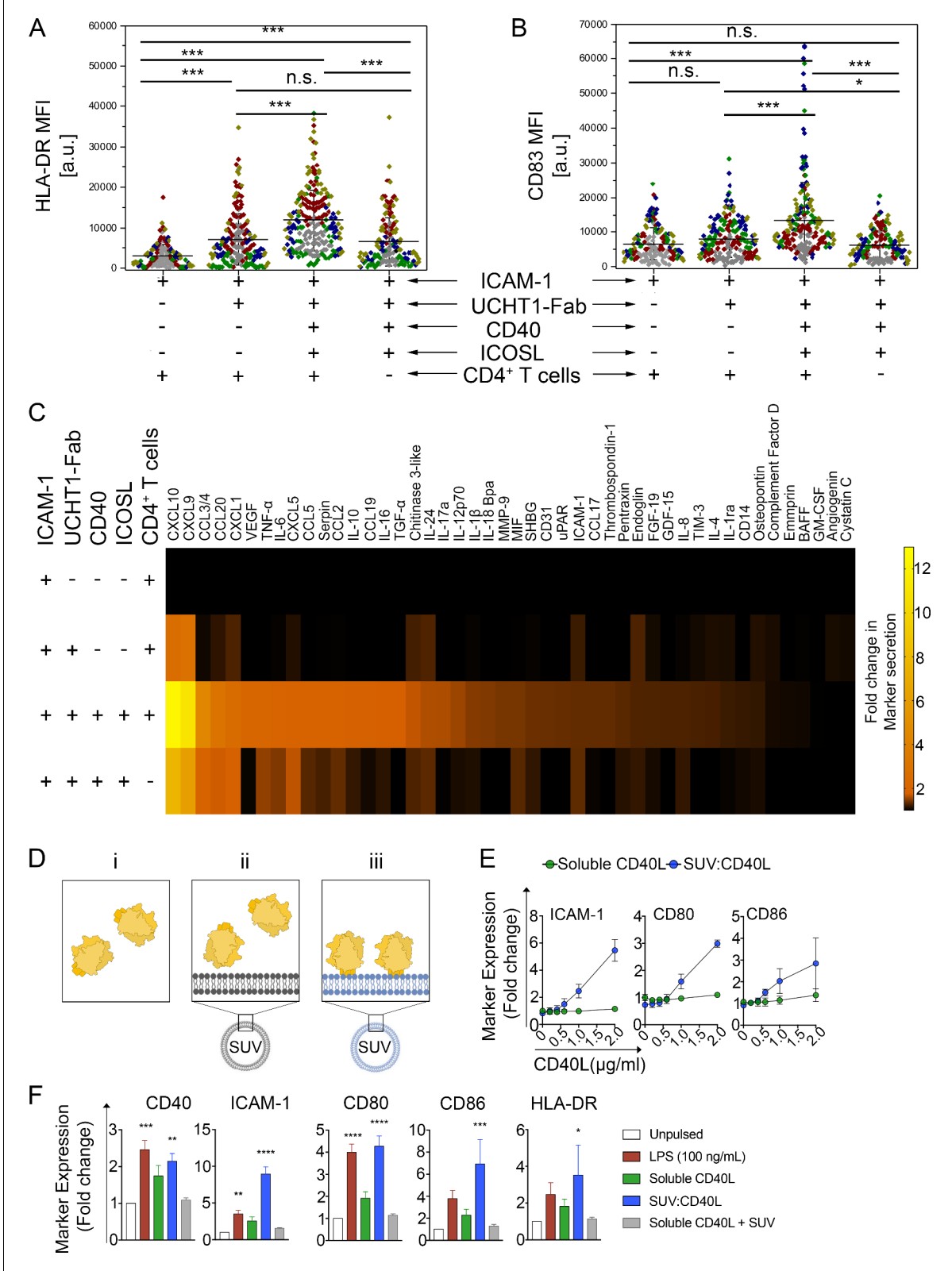

**Figure 8.** CD40L-positive SE left by T cells help DC and high-density vesicular CD40L is sufficient for DC maturation. (A, B) HLA-DR (A) or CD83 (B) expression on the surface of DCs stimulated for 24 hr on PSLB prepared as indicated to present SE. Each symbol represents the mean fluorescent intensity of a cell (n ≥ 20) from five independent donors, which are represented by the different donors. Lines and errors represent means ± SD. ns, not significant; *, p≤0.05; ***, p<0.001; Kruskal-Wallis with Tukey's post-hoc test. (C) After 24 hr of culture, the supernatant of DCs stimulated as in A) and

*Figure 8 continued on next page*

*Figure 8 continued*

B) were analyzed for the presence of different secreted factors. Results from seven independent donors. The fold-change was normalized to the results from the supernatant of DCs incubated with PSLB containing ICAM-1 alone. (D) For assessing the biological significance of presenting CD40L in vesicular structures compared to the proposed biologically active form of soluble trimeric CD40L [i, green], we developed 86 nm diameter synthetic unilamellar vesicles (SUV; *Supplementary file 2D*) using phospholipids either without (ii, gray) or with His-tag, and hence CD40L, binding activity (iii, blue). An equal total mass (Mass eq.) of N-terminal His-tagged recombinant human CD40L was either incubated with DOPC liposomes (mock-SUV [ii, gray]) or used to load 12.5% DOGS-NTA and make vesicular CD40L (SUV:CD40L [iii, green]). As control, a Mass eq. of CD40L was used as soluble, free protein in culture. (E) FCM analyses of moDCs stimulated for 24 hr with either SUVs containing increasing amounts of CD40L monomers (blue; approximately 0, 40, 80, 120, 190 and 230 CD40L molec./SUV respectively) or an equivalent concentration of soluble CD40L (green). Data represent fold change in the GMFIs for ICAM-1, CD80 and CD86 of treated moDCs over unpulsed controls. (F) As in B, moDCs were stimulated using SUV:CD40L at a final load of approximately 230 molec./SUV. An equivalent concentration of soluble CD40L either delivered alone or in combination with SUVs lacking NTA lipids (soluble CD40L + SUV) were used as controls. After 24 hr of stimulation moDCs were collected, FcR blocked, stained and analysed by multicolor FCM for the expression of maturation markers. Given the high variability of arbitrary fluorescence values, the response was normalized as a fold change compared to unpulsed, immature DC controls. Data represent mean + /- SEM collected from five different donors and five independent experiments. One-way ANOVA and Dunn´s multiple comparisons test to the mock treated control was performed. *P* values < 0.05 (*);<0.002 (**); <0.0002 (***);<0.0001 (****) were considered significant.

DOI: https://doi.org/10.7554/eLife.47528.026

The following figure supplement is available for figure 8:

**Figure supplement 1.** SE captured on PSLB efficiently activate moDCs.
DOI: https://doi.org/10.7554/eLife.47528.027

vesicles have CD40L dependent adjuvant-like function when injected into mice (*Sprague et al., 2008*). However, conditional knockout of CD40L in CD4[+] T cells appears to fully account for the classical defects in antibody class switching and T cell help associated with CD40L deficiency (*Horrillo et al., 2014*). Instead, platelet CD40L appears to function as an integrin ligand in blood clotting such that its physiological function is not CD40 dependent, but integrin β3 dependent (*André et al., 2002*). In contrast, T cell SEs are generated in an antigen dependent manner and TCR and CD40L are often present in the same SE, thus adding another level of specificity of this post-T cell product. SE may provide an explanation for reports of antigen specific helper factors (*Guy et al., 1989*). Guy et al. described soluble antigen-specific and MHC-restricted factors that delivered T cell help. Whilst Guy et al suggested a proteolytically cleaved form of TCR that is complexed with other proteins, our study suggests the alternative that SE with TCR and CD40L could account for such an activity.

T cells transfer CD40L to B cells in an agonist pMHC and CD40 dependent manner (*Gardell and Parker, 2017*). We and other investigators speculated that CD40L transfer in extracellular vesicles might be an important component of T cell help (*Gardell and Parker, 2017*; *Dustin, 2014*). Our new results support this speculation in that CD40L is readily incorporated into SE and dSTORM confirms that CD40L and TCR are incorporated in the same CD81 positive vesicles. Gardell and Parker were also able to detect bystander capture of CD40L by CD40 positive, agonist pMHC negative B cells in the presence of CD40 negative, agonist pMHC positive B cells (*Gardell and Parker, 2017*). We were unable to recapitulate 'bystander' transfer of CD40L when TCR agonists and CD40 are presented on different BSLB. This suggests that active processes in the B cells are required to use CD40 to capture CD40L from a T cell without a TCR agonist to form an IS. A recent study suggests that DC also receive CD40L-CD40 signals without cognate pMHC (*Pasqual et al., 2018*).

Both we (*Papa et al., 2017*) and *Gardell and Parker (2017)* utilized anti-CD40L mAb to detect CD40L transfer, which comes with caveats. Since these mAb are function blocking, we naturally were concerned that CD40 in the SLB might compete with the fluorescently tagged anti-CD40L mAb leading to underestimation of transfer. High densities of CD40 in the SLB did suppress the signal with anti-CD40L, but CD40L was still equally depleted from T cells on average, so our measurements of CD40L are likely an underestimate, but over the entire physiological range of CD40 densities we could detect transfer with anti-CD40L mAb. The dSTORM analysis of anti-CD40L localization in SE revealed that CD40L was detected in microclusters distinct from the TCR/ICOS/BST2 microcluster that must correspond to the site of attachment between the SE and the PSLB. This physical segregation of CD40L from the attachment site to the antigen-presenting surface may in part explain its availability to both the anti-CD40L antibody used for detection and CD40 on DCs, which responded functionally to the CD40L on SE. We also noted a CD40/CD40L independent, DC stimulatory activity

associated with released SE, which may be explained by recently reported release of mitochondrial DNA associated with T cell exosomes (*Torralba et al., 2018*), released in parallel with SE.

MS analysis provided a number of new candidates for SE generation and function. We detected many new ESCRT and vesicle trafficking components in SE. We identified two ubiquitin recognizing ESCRT-0 components HRS (Hepatocyte growth factor-regulated tyrosine kinsase substrate; an ESCRT-0 component) and EPN1 (*Shih et al., 2002*), which may explain why we previously found that HRS is not essential for cSMAC formation (*Vardhana et al., 2010*). We previously could find no role for classical ESCRT-II proteins in SE formation (*Vardhana et al., 2010*), but our MS analysis found ALIX, which bridges TSG101 to ESCRT-III components through its Bro-1 domain (*Mahul-Mellier et al., 2006*). Suppression of ALIX had no effect on transfer of TCR and CD40L transfer, suggesting a potential degree of redundancy of ESCRT involved in the release of SE at the IS. Of the candidates investigated by TIRFM, EPN1 was highly concentrated in a ring of discrete punctate structures surrounding the synaptic cleft, which is a pattern observed for other ESCRT components (*Choudhuri et al., 2014*). EPN1 may serve in a sorting complex with AP2, clathrin, EPS15 and dynamin (*Chen et al., 1998*; *Ford et al., 2002*), which were all detected in SE. We do not know if SE that are transferred through the IS can be released from the recipient APC to engage other APC in a TCR dependent or independent manner. But it is clear that SE attached to CD40 bearing SLB does maintain CD40L that is not engaged by CD40. The MS analysis identified ADAM10 in SE. ADAM10 converts CD40 engaged CD40L into soluble trimers (*Yacoub et al., 2013*), which might release the SE bearing non-CD40 engaged clusters of CD40L to allow interaction with other APC. We found that CD40L incorporation into SE and transfer to BSLB was not dependent on ADAM10 dependent shedding, and further suggest that ADAM10 regulates CD40L transfer to APC by shedding CD40L from SE. Further mining of the rich list of proteins from the MS analysis may identify means to selectively interfere with SE formation in vivo to better understand the critical physiological role of CD40L and TCR transfer to APC in synaptic ectosomes.

We can propose the following model for formation of SE containing TCR and CD40L microclusters. TCR-ligand microcluster formation induces signals leading to LFA-1-ICAM-1 dependent spreading and surface exposure of preformed CD40L microclusters (*Video 1* and *Figure 1C,D*). After spreading is complete TCR microclusters translocate towards the IS center with F-actin flow. The CD40L microclusters move randomly in the IS in the absence of CD40 on the APC. In the presence of CD40 on the APC, the CD40L-CD40 microclusters translocate toward the center of the IS in parallel with the TCR-ligand microclusters. The convergence of the TCR-ligand and CD40L-CD40 microclusters at the cSMAC generates opportunities for co-sorting by the ESCRT machinery into membrane buds that give rise to double positive SE, which were observed in just over 50% of events in our dSTORM data. Here, we have demonstrated that the CD40L in the SE is active in inducing DC maturation. The combination of the TCR triggered $Ca^{2+}$ signaling (*Choudhuri et al., 2014*) and the CD40L induced NF-κB activation may generate further antigen specific responses in the APC.

## Materials and methods

### Ethics

Leukapheresis products (non-clinical and de-identified) from donor blood were used as a source of human T cells and monocytes. The Non-Clinical Issue division of National Health Service approved the use of leukapheresis reduction (LRS) chambers products at the University of Oxford (REC 11/H0711/7). Clone 35 was isolated from a healthy volunteer where written informed consent was given. Ethical approval was obtained from the University of Oxford Tropical Ethics Committee (OXTREC).

### T cell lymphoblast, Clone 35 culture and CRISPR gene editing

CD4[+] T cell lymphoblasts were generated from human peripheral blood CD4[+] T cells isolated from healthy donors (*Levine et al., 1997*). Briefly, CD4[+] T cells were isolated by negative selection (RosetteSep Human CD4[+] T cell Enrichment Kit, Stemcell technologies) following the manufacturer's procedure. The CD4[+] T cells were activated for 3 days using anti-CD3/anti-CD28 T-cell activation and expansion beads (Dynabeads, ThermoFisher Scientific) in complete medium (RPMI 1640 media supplemented with 10% heat-inactivated fetal bovine serum, 50 U/ml of Penicillin-Streptomycin, 2 mM L-Glutamine, 10 mM HEPES, 1 mM Sodium Pyruvate, and 100 μM non-essential amino acids) with

100 U/ml of recombinant human IL-2 (PeproTech), which was replaced every 2 days keeping the cells at a concentration of $1.5 \times 10^6$ cells/ml. IL-2 containing media (25 U/mL) was replenished the night before experiments. We refer to these as 'T cells' and they were used on day 10 when all division had ceased. The HLA-DRB1*09:01-restricted T cell clone 35 (specific against the influenza H3 $HA_{338-355}$ peptide NVPEKQTRGIFGAIAGFI) were expanded using at a ratio of 1 clone: two feeder cells (irradiated, pooled PBMCs from 2 to 3 healthy donors) at a total cell concentration of $3 \times 10^6$ cells/ml in RPMI 1640 supplemented with 10% heat-inactivated AB human serum and 30 µg/ml of PHA for three days. Then, 100 U/ml of recombinant human IL-2 were added to fresh media, which was replaced every 2 days. For CRISPR gene editing experiments, $CD4^+$ T cells were activated with anti-CD3/anti-CD28 beads as described above. After 3 days activation beads were removed and cells washed three times in Opti-MEM (Gibco). *Trans*-activating Crispr RNA (Alt-R tracrRNA) and either target or control (CD19) Alt-R CRISPR-Cas9t gRNA were obtained from IDT. For $1.5 \times 10^6$ T cells Alt-R tracrRNA and Alt-R CRISPR-Cas9 gRNA were mixed in equimolar amounts (150 pmol) prior to incubation at 95℃ for 5 min and resultant duplex allowed to cool to room temperature. 150 pmol of ALT-R S.p Cas9 Nuclease V3 (IDT) and duplexed gRNA were mixed in IDT nuclease-free duplex buffer and assembled for 15 min at 37℃. ALT-RCas9 Electroporation Enhancer (IDT) was added (150 pmol) to the resultant ribonucleoprotein and added to $1.5 \times 10^6$ T cells in 50 µl of Opti-MEM prior to electroporation in an ECM 880 Electro Square Porator (BTX Harvard Apparatus). The cells were expanded for 4 days in recombinant human IL-2 supplemented RPMI as described above.

## MoDC culture and activation

Monocyte-derived dendritic cells (moDC) were generated from the peripheral blood of healthy adults by first isolating monocytes by negative selection (RosetteSep Human Monocyte Enrichment Cocktail, STEMCELL technologies) following the manufacturer's procedure. Then, $1.5–2 \times 10^6$ monocytes/ml/cm$^2$ were stimulated in complete media supplemented with 100 ng/ml of recombinant human GM-CSF and 200 ng/ml of recombinant human IL-4 (*Sallusto and Lanzavecchia, 1994*). After 5 to 7 days of culture, moDC were used in experiments.

## Antibodies

Primary monoclonal antibodies (mAb) used for dSTORM were anti-TCR-Alexa Fluor (AF) 488 (clone IP26; BioLegend), anti-CD40L-AF647 (clone 24–31; BioLegend), anti-ICOS-AF647 (clone C398.4A; BioLegend), anti-BST2-AF647 (clone RS38E; BioLegend), anti-HLA-DR-AF488 (clone L243; BioLegend), anti-CD81-AF647 (clone 5A6; BioLegend) anti-CD83-AF647 (clone HB15e; BioLegend) and Wheat Germ Agglutinin WGA-CF568 (Biotum) to label the surface of the SEs. All antibody clones used to assess relative or absolute quantification of protein transfer from cells to BSLB are listed in *Supplementary file 2A*. Isotype controls matching the relevant fluorescent dyes were used for background correction and gating. Other mAb or affinity purified antibodies are described with specific methods below.

## Small unilamellar vesicles (SUVs)

SUV are defined as vesicles in the 20–100 nm range. SUV were formed by extrusion as described using the Avanti Miniextruder with a 100 nm filter (*Crites et al., 2015*). When SUV were used to mimic SE, all lipids were combined prior to SUV formation, whereas BSLB and PSLB composition could be determined by mixing different proportions of stock SUVs as the final bilayer composition is determined by the average of the input SUV. NTA-SUVs for attachment of His tagged proteins were composed of 85.5 mol% DOPC, 2 mol% head group labeled ATTO-390-DOPE, and 12.5 mol% DOGS-NTA at a total lipid concentration of 4 mM. Plain SUVs that were not able to bind His tagged proteins, were composed of 98 mol% DOPC and 2% ATTO-390-DOPE at a total lipid concentration of 4 mM. Stock SUV for formation of BSLB or PSLB were composed of 0.4 mM solution of lipids in PBS with 100 mol% DOPC; 75 mol% DOPC and 25 mol% DOGS-NTA; 98 mol% DOPC and 2 mol% DOPE-CAP-Biotin; or 98% DOPC; 2 mol% ATTO-(390 or 488)-DOPE. These stocks could be mixed in different ratios prior to formation of BSLB or PSLB to generate mobile bilayers of the desired final composition. All lipids were purchased from Avanti Polar Lipids, Inc (Alabaster, AL).

## Nanoparticle Tracking Analysis

A 10 μL aliquot of SUVs or eluted SE preparation was re-suspended in PBS in a 1:100 dilution and kept on ice for Nanoparticle Tracking Analysis (NTA). The instrument used for NTA was Nanosight NS300 (Malvern Instruments Ltd) set on light scattering mode and instrument sensitivity of 15. Measurements were taken with the aid of a syringe pump to improve reproducibility. Three sequential recordings of 60 s each were obtained per sample and NTA 3.2 software was used to process and average the three recordings to determine the mean size. moDC activation by CD40L on SUV.

His-tagged recombinant soluble CD40L (sCD40L, BioLegend) was incubated with NTA-SUV or plain SUV, at ratios designed to match CD40L densities found on SE for 20 min at 24°C prior to addition to the moDCs. After 24 hr, moDCs were recovered by spinning down plates at 1500 rpm for 5 min and resuspended in flow cytometry staining buffer (10% Heat-Inactivated Goat Serum, 0.04% sodium azide in PBS pH 7.4) and incubated for 30 min at 4°C. A final concentration of 10–30 nM of each mAb was used. The multicolor panel included anti-HLA-DR PerCP (clone L243), anti-CD40 AF647 (clone 5C3), anti-ICAM-1 Brilliant Violet 510/Brilliant Violet 785 (clone HA58), anti-CD80 PE (clone 2D10), anti-CD86 Brilliant Violet 785 (clone IT2.2) and anti-ICOSL PE-Cy7 (clone 2D3). Isotype control antibodies clones MOPC-21 (IgG1, κ), MOPC-173 (IgG2a, κ) and MPC-11 (IgG2b, κ) were used matching the relevant fluorescent dyes. Staining was performed for 30 min at 4°C in the dark and constant agitation after which cells were washed twice and single cell fluorescence measurements were made by flow cytometry.

## Bead Supported Lipid Bilayers

Silica beads (5.0 μm diameter, Bangs Laboratories, Inc) were washed extensively with PBS in a 1.5 ml conical microcentrifuge tubes. BSLBs were formed by incubation with mixtures of SUVs to generate a final lipid composition of 0.2 mol% ATTO 488-DOPE; 12.5 mol% DOGS -NTA and a mol% of DOPE-CAP-Biotin to yield 10–5000 molecules/$\mu m^2$ UCHT1-Fab in DOPC at a total lipid concentration of 0.4 mM. The resultant BSLB were washed with 1% human serum albumin (HSA)-supplemented HEPES-buffered saline (HBS), subsequently referred to as HBS/HSA. After blocking with 5% casein in PBS containing 100 μM NiSO$_4$, to saturate NTA sites, 50 μg/mL unlabelled streptavidin was then coupled to biotin head groups by incubation with concentrations of streptavidin determined to yield 10–5,000 molec. /$\mu m^2$ site densities. After 20 min, the BSLB were washed 2x with HBS-HSA and biotinylated UCHT1-Fab (variable density as indicated), His-tagged ICAM-1 (200 molec. /$\mu m^2$), CD40 (500 molec./$\mu m^2$), and ICOSL (100 molec./$\mu m^2$) were then incubated with the bilayers at concentrations to achieve the indicated site densities (in range of 1–100 nM). Excess proteins were removed by washing with HBS/HSA after 20 min. T cells (5 × 10$^5$/well) were incubated with BSLB at 1:1 ratio in a V-bottomed 96 well plate (Corning) for 1 hr at 37°C in 100 μl HBS/HSA. BSLB: cell conjugates were pelleted at 500 x g for 1 min prior to resuspension in 50 mM EDTA in PBS at 4°C to release His-tagged proteins from the BSLB, while leaving the UCHT1-Fab attached, thus selectively retaining TCR$^+$ SE. The single BSLB and cells were gently resuspended prior to staining for flow cytometry analysis or sorting.

## Calibration of flow cytometry data

T cells and BSLB were analyzed using antibodies with known AF647:Ab ratio (*Supplementary file 2A*) in parallel with the Quantum AF647 Molecules of Equivalent Soluble Fluorescent dye (MESF) beads, allowing the calculation of the absolute number of mAb bound per T cell and per BSLB after subtraction of unspecific signals given by isotype control antibodies.

## Airyscan microscopy

Airyscan imaging of BSLB-cell conjugates was performed on a confocal laser-scanning microscope Zeiss LSM 880 equipped with Airyscan detection module (Zeiss, Oberkochen, Germany) using the Plan-Apochromat 63×/1.46 Oil objective (Zeiss, Oberkochen, Germany). The Argon laser at 488 nm and diode laser at 561 nm were used as excitation sources, with power setting of ∼ 1% and∼6%, respectively, which is equivalent to 1 mW and 10 mW. The powers were set in this range in order to achieve the comparable strength of fluorescent signal for both channels. Fluorescence emission was collected at around 515 nm and 653 nm for the green and magenta channels, respectively, with the following filters BP420-480+BP495-550 (green) and BP555-620+LP645 (magenta). The emission

signals were collected on the 32 channel GaAsP-PMT Airy detector. The datasets were acquired as Z-stacks with 43.5 nm pixel size and 185 nm axial steps, which correspond to ~ 50–55 slices per 3D data set. ZEN Airyscan software (Zeiss) was used to process the acquired data sets. This software processes each of the 32 Airy detector channels separately by performing filtering, deconvolution and pixel reassignment in order to obtain images with enhanced resolution and improved signal to noise ratio. The value of Wiener filter in ZEN software was chosen in accordance with the value in 'auto' reconstruction modality and was set around 7, to ensure the absence of deconvolution arte-facts (*Korobchevskaya et al., 2017*). Drift was corrected using the MultiStackReg plug-in of ImageJ (National Institute of Health). Rendering was performed in Imaris software (Bitplane).

## Planar Supported Lipid Bilayers (PSLB)

SUV mixtures were injected into flow chambers formed by sealing acid piranha cleaned glass cover-slips to adhesive backed plastic manifolds with six flow channels (StickySlide VI 0.4; Ibidi) (*Papa et al., 2017*). After 20 min the channels were flushed with HBS-HSA without introducing air bubbles to remove excess SUVs. After blocking for 20 min with 5% casein supplemented with 100 µM $NiCl_2$, to saturate NTA sites, followed by 15 min incubation with streptavidin (Sigma Aldrich), washing and then monobiotinyated or His-tagged proteins were incubated on bilayers for additional 20 min. Protein concentrations required to achieve desired densities on bilayers were calculated from calibration curves constructed from flow-cytometric measurements of BSLB, compared with reference beads containing known numbers of the appropriate fluorescent dyes (Bangs Laboratories). Bilayers were continuous liquid disordered phase as determined by fluorescence recovery after photobleaching with a 10 µm bleach spot on an FV1200 confocal microscope (Olympus).

## T cell immunological synapse formation on PSLB

$CD4^+$ T cells were incubated at 37°C on SLB containing either ICAM-1 alone, ICAM-1 and UCHT1-Fab or ICAM-1 UCHT1-Fab, CD40 and ICOSL. After 20–90 min of incubation the cells either fixed with 4% electron microscopy grade formaldehyde in PHEM buffer (10 mM EGTA, 2 mM $MgCl_2$, 60 mM Pipes, 25 mM HEPES, pH 7.0), permeabilized with 0.1% Triton X-100 (if necessary for access to intracellular spaces) and stained with primary conjugated antibodies and imaged. Alternatively, the cells were washed off with cold PBS and the SE left behind were stained with directly conjugated antibodies and fixed with 4% formaldehyde in PHEM buffer. Prior the labeling of moDCs with mAbs on SLB for TIRF imaging, the cells were blocked for Fc receptors with 5% HSA and 5% goat or donkey serum for 1 hr at 24°C.

## Total internal reflection fluorescence microscopy (TIRFM)

TIRFM was performed on an Olympus IX83 inverted microscope equipped with a 4-line (405 nm, 488 nm, 561 nm, and 640 nm laser) illumination system. The system was fitted with an Olympus UApON 150 × 1.45 numerical aperture objective, and a Photomertrics Evolve delta EMCCD camera to provide Nyquist sampling. Live experiments were performed with an incubator box maintaining 37°C and a continuous autofocus mechanism. Quantification of fluorescence intensity was performed with ImageJ (National Institute of Health). dSTORM imaging and data analysis.

For three colour dSTORM imaging extracellular vesicles were stained using either wheat germ agglutinin (WGA) directly conjugated with CF568 (Biotium) or anti-CD81-AF647. First, 640 nm laser light was used for exciting the AF647 dye and switching it to the dark state. Second, 488 nm laser light was used for exciting the AF488 dye and switching it to the dark state. Third, 560 nm laser light was used for exciting the CF568 dye and switching it to the dark state. An additional 405 nm laser light was used for reactivating the AF647, AF488 and CF568 fluorescence. The emitted light from all dyes was collected by the same objective and imaged onto the electron-multiplying charge-coupled device camera with an effective exposure time of 10 ms. A maximum of 5000 frames for antibodies conjugated with AF647, CF568 and AF488 condition were acquired. For visualizing the WGA labelled extracellular vesicles minimum of 80,000 frames were acquired. For each receptor, the specificity of the labeling was confirmed by staining the vesicles with isotype-matched control antibodies (data not shown).

Because multicolour dSTORM imaging is performed in sequential mode by using three different optical detection paths (same dichroic but different emission filters), an image registration is

required to generate the final three-color dSTORM image (*Bálint et al., 2013*; *Bates et al., 2012*; *Lopes et al., 2017*). Therefore, fiducial markers (TetraSpek Fluorescent Microspheres; Invitrogen) of 100 nm, which were visible in 488 nm, 561 nm and 640 nm channels, were used to align the 488 nm channel to 640 nm channel. The difference between 561 nm channel and 640 nm channel was negligible and therefore transformation was not performed for 561 nm channel. The images of the beads in both channels were used to calculate a polynomial transformation function that maps the 488 nm channel onto the 640 nm channel, using the MultiStackReg plug-in of ImageJ (National Institute of Health) to account for differences in magnification and rotation, for example. The transformation was applied to each frame of the 488 nm channel. dSTORM images were analyzed and rendered as previously described (*Bates et al., 2007*; *Huang et al., 2008*) using custom-written software (Insight3, provided by B. Huang, University of California, San Francisco). In brief, peaks in single-molecule images were identified based on a threshold and fit to a simple Gaussian to determine the x and y positions. Only localizations with photon count $\geq$ 2000 photons were included, and localizations that appeared within one pixel in five consecutive frames were merged together and fitted as one localization. The final images were rendered by representing the x and y positions of the localizations as a Gaussian with a width that corresponds to the determined localization precision. Sample drift during acquisition was calculated and subtracted by reconstructing dSTORM images from subsets of frames (500 frames) and correlating these images to a reference frame (the initial time segment). ImageJ was used to merge rendered high-resolution images (National Institute of Health).

## CBC analysis

Coordinate-based colocalization (CBC) mediated analysis between two proteins was performed using an ImageJ (National Institute of Health) plug-in (*Ovesný et al., 2014*) based on an algorithm described previously (*Malkusch et al., 2012*). To assess the correlation function for each localization, the x-y coordinate list from 488 nm and 640 nm dSTORM channels was used. For each localization from the 488 nm channel, the correlation function to each localization from the 640 nm channel was calculated. This parameter can vary from −1 (perfectly segregated) to 0 (uncorrelated distributions) to +1 (perfectly colocalized). The correlation coefficients were plotted as a histogram of occurrences with a 0.1 binning. The Nearest-neighbor distance (NND) between each localization from the 488 nm channel and its closest localization from the 640 nm channel was measured and plotted as the median NND between localizations per cell.

## Cross-correlation analysis

Cross correlation analysis is independent of the number of localizations and is not susceptible to over-counting artifacts related to fluorescent dye re-blinking and the complements other approaches (*Stone et al., 2017*). Cross-correlation analysis between two proteins was performed using MATLAB software provided by Sarah Shelby and Sarah Veatch from University of Michigan. Regions containing cells were masked by region of interest and the cross-correlation function from x-y coordinate list from 488 nm and 640 nm dSTORM channels was computed from these regions using an algorithm described previously (*Stone et al., 2017*; *Shelby et al., 2013*; *Veatch et al., 2012*). Cross-correlation functions, C(r,q), were firstly tabulated by computing the distances between pairs of localized molecules, then C(r) is obtained by averaging over angles. Generally, C(r) is tabulated from ungrouped images, meaning that localizations detected within a small radius in sequential frames are counted independently. Finally, a normalized histogram with these distances was constructed into discrete bins covering radial distances up to 1000 nm. Cross-correlation functions only indicate significant correlations when the spatial distribution of the first probe influences the spatial distribution of the second probe, even when one or both of the probes are clustered themselves. Error bars are estimated using the variance within the radial average of the two dimensional C(r, q), the average lateral resolution of the measurement, and the numbers of probes imaged in each channel. The cross-correlation function tabulated from the images indicates that molecules are highly colocalized, where the magnitude of the cross-correlation yield (C(r)>1) is higher than randomly co-distributed molecules (C(r)=1).

## Cytokine array

Primary moDCs were incubated on the extracellular vesicles der(ived from CD4$^+$ T cells on SLB containing either only ICAM-1 (200 molec/μm$^2$), ICAM-1 and UCHT1-Fab (300 molec/μm$^2$), or ICAM-1, UCHT1-Fab, CD40 (500 molec/μm$^2$) and ICOSL (100 molec/μm$^2$), at 37°C for 24 hr. Cell supernatants were recovered and centrifuged at 350 g for 5 min at RT to remove cells and cell debris. Cytokine production was quantified in the supernatants by Human XL Cytokine Array kit (ARY022B; R and D Systems), according to manufacturer's instructions. The positive signal from cytokines was determined by measuring the average signal of the pair of duplicate spots by using ImageJ (National Institute of Health). Differences between arrays were corrected by using the average intensity of positive spots within the array. Fold change of the cytokine production between conditions was determined by normalizing the data to SLB containing only ICAM-1.

## Mass Spectrometry

AF488+ BSLB were sorted on a FACS ARIA III and lysed by sonication (Bioruptor Pico) in 0.5% NP-40 in 50 mM ammonium bicarbonate and 6 M urea. Cysteines were reduced and alkylated by addition of first 5 μl of 200 mM dithiothreitol (30 min at 24°C) and 10 μl of 200 mM iodoacetamide (60 min at RT in dark). The protein solution was then precipitated with chloroform and methanol (*Wessel and Flügge, 1984*), and resuspended in 6 M Urea. For digest the protein solution was diluted in 50 mM ammonium bicarbonate, pH 7, and 0.6 μg trypsin was added for digest at 37°C overnight. Peptides were desalted with a C18 solid phase extraction cartridge (SOLA, Thermo Fisher Scientific) and resuspended in 15 μl 2% acetonitrile and 0.1% trifluoroacetic acid in water. Samples were analyzed on a LC-MS/MS platform consisting of Orbitrap Fusion Lumos coupled to a UPLC ultimate 3000 RSLCnano (both Thermo Fisher Scientific). Samples were loaded in 1% acetonitrile and 0.1% trifluoroacetic acid in water and eluted with a gradient from 2% to 35% acetonitrile, 0.1% formic acid and 5% dimethylsulfoxide in water in 60 min with a flow rate of 250 nl/min on an EASY-Spray column (ES803, Thermo Fisher Scientific). The survey scan was acquired at a resolution of 120.000 between 380–1500 m/z and an automatic gain control target of 4E5. Selected precursor ions were isolated in the quadrupole with a mass isolation window of 1.6 Th and analyzed after CID fragmentation at 35% normalized collision energy in the linear ion trap in rapid scan mode. The duty cycle was fixed at 3 s with a maximum injection time of 300 ms, AGC target of 4000 and parallelization enabled. Selected precursor masses were excluded for the following 60 s. Proteomic data was analyzed in Maxquant (V1.5.7.4, ref) using default parameters and Label Free Quantitation. The data was searched against the mouse canonical Uniprot database (29/07/2015) and the human Uniprot database (15/10/2014). FDR on peptide and protein level were set to 1%. Second peptide and 'match between runs' options were enabled.

The mass spectrometry proteomics data have been deposited to the ProteomeXchange Consortium via the PRIDE (*Vizcaíno et al., 2016*) partner repository with the dataset identifier PXD007988 (https://www.ebi.ac.uk/pride/archive/projects/PXD007988).

## Statistical analysis

All statistical analyses were performed using SigmaPlot 13.0 (Systat Software Inc), OriginPro 2017 software (OriginLab) or GraphPad Prism v 7.0 and 8.0 (GraphPad Software, Inc). Statistical analyses are detailed in each figure legend.

## Acknowledgements

We thank E Kurz, A Afrose, H Rada and P Hernandez-Varas for technical assistance; A M Santos, S J Davis, J Felce, S Campion, P Borrow and P Lipsky for helpful suggestions; and P Borrow, C Vinuesa and G Victora for critical reading. Supported by the ERC AdG 670930 (MLD), the Wellcome Trust PRF 100262 (MLD), the Kennedy Trust (to MLD and BMK), the NIH AI043542 (MLD), the NIH tetramer core facility, the EMBO ALTF 1420–2015 (PFCD), Medical Research Council (YCD and TD), UK and Chinese Academy of Medical Sciences Innovation Fund 2018-I2M-2–002 (YCD and TD), Cancer Research UK A19277 (EO) and the Research Council of Norway in conjunction with Marie Sklodowska-Curie Actions Project number 275466 (AK).

## Additional information

### Competing interests

Michael L Dustin: Reviewing editor, *eLife*. The other authors declare that no competing interests exist.

### Funding

| Funder | Grant reference number | Author |
|---|---|---|
| European Commission | AdG 670930 | David G Saliba<br>Pablo F Céspedes-Donoso<br>Štefan Bálint<br>Ewoud B Compeer<br>Michael L Dustin |
| Wellcome | PRF 100262 | Michael L Dustin |
| Cancer Research UK | UK A19277 | Eric O'Neill |
| Chinese Academy of Sciences | 2018-I2M-2-002 | Yanchun Peng<br>Tao Dong |
| National Institutes of Health | AI043542 | Michael L Dustin |
| Kennedy Trust for Rheumatology Research | | Michael L Dustin<br>Benedikt M Kessler |
| European Molecular Biology Organization | ALTF 1420-2015 (in conjunction with the European Commission (LTFCOFUND2013, GA-2013-609409) and Marie Sklodowska-Curie Actions). | Pablo F Céspedes-Donoso |
| Research Council of Norway | The Research Council of Norway in conjunction with Marie Sklodowska-Curie Actions 275466 | Audun Kvalvaag |
| H2020 Marie Skłodowska-Curie Actions | The Research Council of Norway in conjunction with Marie Sklodowska-Curie Actions 275466 | Audun Kvalvaag |

The funders had no role in study design, data collection and interpretation, or the decision to submit the work for publication.

### Author contributions

David G Saliba, Pablo F Céspedes-Donoso, Štefan Bálint, Conceptualization, Data curation, Formal analysis, Validation, Investigation, Methodology, Writing—original draft, Writing—review and editing; Ewoud B Compeer, Data curation, Formal analysis, Validation; Kseniya Korobchevskaya, Formal analysis; Salvatore Valvo, Resources, Validation, Methodology; Viveka Mayya, Yanchun Peng, Tao Dong, Maria-Laura Tognoli, Eric O'Neill, Sarah Bonham, Benedikt M Kessler, Resources; Audun Kvalvaag, Methodology; Roman Fischer, Resources, Data curation; Michael L Dustin, Conceptualization, Resources, Supervision, Funding acquisition, Visualization, Writing—original draft, Project administration, Writing—review and editing

### Author ORCIDs

David G Saliba (iD) https://orcid.org/0000-0001-9523-0586
Pablo F Céspedes-Donoso (iD) https://orcid.org/0000-0002-1641-4107
Štefan Bálint (iD) http://orcid.org/0000-0003-4470-5881
Ewoud B Compeer (iD) http://orcid.org/0000-0002-3050-7633
Tao Dong (iD) http://orcid.org/0000-0003-3545-3758
Roman Fischer (iD) http://orcid.org/0000-0002-9715-5951
Michael L Dustin (iD) https://orcid.org/0000-0003-4983-6389

### Ethics

Human subjects: Leukapheresis products (non-clinical and de-identified) from donor blood were used as a source of human T cells and monocytes. The Non-Clinical Issue division of National Health Service approved the use of leukapheresis reduction (LRS) chambers products at the University of Oxford (REC 11/H0711/7). Clone 35 was isolated from a healthy volunteer where written informed consent was given. Ethical approval was obtained from the University of Oxford Tropical Ethics Committee (OXTREC).

### Decision letter and Author response

Decision letter https://doi.org/10.7554/eLife.47528.034
Author response https://doi.org/10.7554/eLife.47528.035

---

## Additional files

### Supplementary files

• Supplementary file 1. Legend. Proteins Identified by MS/MS analysis and represented in *Figure 5*. Fold and Log Fold two represent fold change of BSLB coated with ICAM-1, UCHT1-Fab, CD40, ICOSL over BSLB coated with ICAM-1, CD40 and ICOSL. Colors represent proteins are found for particular pathways than is predicted by chance as determined by reactome pathway
DOI: https://doi.org/10.7554/eLife.47528.028

• Supplementary file 2. Supplementary tables. (**A**) Summary of antibodies used for determination of relative and absolute enrichment of bead-transferred proteins. (**B**) Surface CD40L levels and densities on T cells exposed to different BSLB substrates (either No UCHT1-Fab ± CD40 or UCHT1-Fab ± CD40). (**C**) Estimated number of CD40L molecules per SE and estimated CD40L densities on SE using parameters obtained by dSTORM and quantitative FCM. (**D**) Size distribution of SUVs as determined by Nanoparticle Tracking Analyses using light scattering and Brownian motion.
DOI: https://doi.org/10.7554/eLife.47528.029

• Transparent reporting form
DOI: https://doi.org/10.7554/eLife.47528.030

### Data availability

The mass spectrometry proteomics data have been deposited to the ProteomeXchange Consortium via the PRIDE (Vizcaíno et al 2016) partner repository with the dataset identifier PXD007988 (https://www.ebi.ac.uk/pride/archive/projects/PXD007988).

The following dataset was generated:

| Author(s) | Year | Dataset title | Dataset URL | Database and Identifier |
|---|---|---|---|---|
| Saliba DG, Cespedes-Donoso PF, Balint S, Roman Fischer, Benedikt M Kessler, Michael L Dustin | 2019 | Synaptic Ectosome Proteome | https://www.ebi.ac.uk/pride/archive/projects/PXD007988 | PRIDE, PXD007988 |

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
