## [Decision Letter]

Thank you for submitting your article "Composition and structure of synaptic ectosomes exporting antigen receptor linked to functional CD40L from T_H_ cells" for consideration by *eLife*. Your article has been reviewed by three peer reviewers, and the evaluation has been overseen by a Reviewing Editor and Arup Chakraborty as the Senior Editor. The following individuals involved in review of your submission have agreed to reveal their identity: Pavel Tolar (Reviewer #1); Morgan Huse (Reviewer #2); Francisco Sánchez Madrid (Reviewer #3).

The reviewers have discussed the reviews with one another and the Reviewing Editor has drafted this decision to help you prepare a revised submission.

As you will see from the comments below, the reviewers are excited about the work. However, they all agree that the main point that needs to be proven to strengthen the paper is to demonstrate that indeed ectosomes are involved. The way to do this, we think, is to genetically interfere with well-known proteins responsible for their secretion. We are copying the reviewers' unvarnished comments below, which might help provide suggestions as to how to approach this point, and also address minor points.

*Reviewer #1:*

This paper investigates the release of ectosomes from helper T cells. Previously the authors showed that T cells release their TCRs into synaptic ectosomes in an ESCRT-dependent manner. Here, the paper looks at the broader composition of the ectosomes, and investigate their role in delivering CD40L and ICOS to APCs. The authors recapitulate older studies showing a TCR-stimulated delivery of CD40L to the synapse and extend these findings to show that CD40L (and ICOS) are also sorted into ectosomes. The ectosome material is functional in stimulating CD40 on dendritic cells. The CD40-CD40L signaling in the synapse is still poorly understood and the ectosomal mechanism could provide novel insights.

Unfortunately, the paper does not solve critical issues about the mechanism of CD40L inclusion into the ectosomes and whether such inclusion truly regulates CD40 signaling. There is also a number of small issues with the figures that make reading and interpreting the data difficult.

Major issues:

1) What is the mechanism of CD40L transfer onto ectosomes? There is limited colocalization of CD40L with TCR in the synapse and on ectosomes, suggesting the mechanism may be distinct from that of the TCR. Without a specific mechanism identified, it is difficult to exclude that a large fraction of the CD40L seen on the SPLBs is just CD40L ripped off the T cells by CD40 during samples preparation. This could have happened in all the experiments and it would still be "specific" as it is pulled by CD40. Even in the kinapse video, the cell seems to be attached to the old synapse site by a membrane tether that is seen stretching and most of the CD40 dissipates as the cell moves away. Thus, the majority of CD40L was still in the T cell membrane, not in ectosomes. Similarly, the proteomics captures a number of endocytic and ESCRT proteins that should remain in the cells when producing ectosomes, suggesting some membrane rupture occurred and the ligands in the SPLB pulled these complexes along. Working towards the mechanism, Figure 4 suggests that CD40L transfer to the SPLB beads is inhibited by MG132, but the inhibition is only partial and MGT132 has many effects beyond reducing free ubiquitin. In the same experiments the authors conclude that TCR transfer is not inhibited by MG132, yet there is a similar decrease in transfer as for the CD40L. This will need to be explained.

2) What is the role of CD40 and ICOSL in mediating CD40L and ICOS release into ectosomes? The authors show that CD40L (or ICOS) accumulates in the synapse and in ectosomes strictly in correlation with the concentration of the CD40 (ICOSL) and independently of the concentration of antigen. Since it is possible that without the capture of CD40L by CD40, no accumulation would be seen simply because the CD40L is either internalised or secreted without capture, it is not clear what the steps are leading to inclusion of CD40L in the ectosomes and how does TCR regulate this process. Is it direct vesicle release, or plasma membrane incorporation followed by proteolytic cleavage or extrusion? In Figure 4, the authors show a depletion of CD40L in T cells that interacted with antigen and CD40, but not in cells that interacted only with antigen. However, it is not clear if the staining shows cell surface or total CD40L levels.

3) What is the role CD40L inclusion into ectosomes? It is known that CD40L works better if incorporated on particles or surfaces, however, this is already accomplished in the synapse. As the authors show, all the CD40L detectable in the synapse is captured via CD40, so CD40 signaling is already activated before ectosome release. After ectosome release, the ectosomes are bound to CD40 so they cannot amplify the signaling or spread to other APCs. Is the duration of CD40 signaling influenced?

*Reviewer #2:*

"Composition and structure of synaptic ectosomes exporting antigen receptor linked to functional CD40L from T_H_ cells", by Saliba et al., applies biochemical, proteomic, and imaging approaches to explore the requirements for protein sorting into synaptic ectosomes (SEs), the organization of SEs, and the role of SEs during T cell effector responses. In general, I found the manuscript to be interesting and informative, and I also thought that the conclusions were well supported by the data presented. I have just a few comments (enumerated below), which should be well within the capabilities of the authors to address.

1) Although the constellation of proteins enriched during the authors' BSLB-based purification strategy (e.g. Figure 2) does suggest that they are isolating SEs, I am not entirely convinced that the approach isn't just ripping off pieces of synaptic membrane indiscriminately. The Dustin lab has demonstrated quite nicely in the past that TSG101 and VPS4 are required for SE formation. They should demonstrate here that both proteins are required for CD40L and ICOS enrichment in their putative BSLB-based SE preparations.

2) In subsection “Mass Spectrometry (MS) of SEs reveals enrichment of ESCRT proteins and TCR signaling”, the authors write "We allowed 50 million CD4^+^ effector T cells to form IS on BSLB with ICAM-1, CD40, ICOSL in the presence or absence of UCHT1-Fab, disengaged the T cells and BSLB by incubation with ice-cold PBS/EDTA, sorted 5 million SE and subjected them to MS analysis after extraction of lipids." Do the authors mean "5 million BSLBs"? If they really do mean "SE", how did they dissociate SEs from BSLBs in their protocol?

3) The polarization of CD40L shown in Figure 1C should be quantitated over multiple cells.

4) In subsection “Selective transfer of CD40L and ICOS into SE”, the authors write "This ICOSL driven TCR independent synapse may exert some control over migration of T cells, but it did not lead to CD40L transfer in any setting and thus does not appear to directly elicit delivery of T cell help." What is the basis for this statement? Did the authors measure antigen independent, ICOSL dependent help?

5) In the Discussion section the authors write "It doesn't escape our attention that SE may provide an explanation for reports of antigen specific helper factors (43)." They should elaborate.

*Reviewer #3:*

The work by Saliba et al. addresses the impact of ectosomes formed at the T cell membrane during the formation of transient immune synapses with putative antigen presenting cells in the biology of dendritic cells upon the uptake of these extracellular vesicles. The authors use different methodologies and approaches to determine differential constituents, with a focus on CD40L, participating in the production of ectosomes/extracellular vesicles by donor cells. This is a relevant process which still requires further research, although already addressed by authors and other groups in terms of determining the ability of T cells of producing and secreting different kinds of extracellular vesicles, and their impact on contacting, nearby or distant recipient cells. The work is well executed and discussed and it is pertinent to the field of cell communication and signaling.

A main question that is relevant and seems to be sidestepped throughout the manuscript, although somehow discussed, is how to differentiate between different kinds of extracellular vesicles that may form part of the T cell secretome released into the immune synapse. Authors may further address that the observed particles are only ectosomes and assess, or at least discuss, other sources for vesicles. A possible mechanism for ectosome formation may rely on the presence of particular proteins. Tetraspanins seem to be common, as well as some components of the ESCRT complexes. However, Epsin 1 protein, a well-known adaptor for AP2 and clathrin, but not known in the formation of intraluminal vesicles as components of multivesicular bodies (e.g.) is also there. If authors could demonstrate that the protein Epsin 1 is present at the plasma membrane, and contributes to the formation of ectosomes, and maybe not to the biogenesis of intraluminal vesicles, this would be a piece of evidence very supportive. The average size and number of detected TCR microclusters is similar to that of analyzed vesicles, which would support such a process.

---

## [Author Response]

Reviewer #1:Major issues:1) What is the mechanism of CD40L transfer onto ectosomes? There is limited colocalization of CD40L with TCR in the synapse and on ectosomes, suggesting the mechanism may be distinct from that of the TCR. Without a specific mechanism identified, it is difficult to exclude that a large fraction of the CD40L seen on the SPLBs is just CD40L ripped off the T cells by CD40 during samples preparation.

The super-resolution dSTORM imaging is important in the conclusions of the study. We were also confused by the obvious displacement of the centers of the UCHT1-Fab puncta and CD40L puncta and modest PCC scores in the conventional TIRFM images of cSMACs and patches of vesicles. The technically demanding 3 color dSTORM imaging really transformed the way we looked at this and while it doesn’t reveal the mechanism, it demonstrates that TCR microclusters come together with CD40L microclusters in single vesicle in about half of the events. We can now show that the CD40L inclusion in the SE is even more sensitive to ESCRT (TSG101 and VPS4b) than the TCR based on new CRISPR gene editing in primary cells (subsection “Selective transfer of CD40L and ICOS into SE” and new Figure 2D, E and Figure 2—figure supplement 1B). Perturbation of other ESCRT components identified by mass spec also reduced transfer of CD40L and CD81 to SE (subsection “Mass Spectrometry (MS) of SEs reveals enrichment of ESCRT proteins and TCR signaling.” and new Figure 5F and Figure 5—figure supplement 2). These results show that even for strong interactions like UCHT1-Fab-TCR interaction that there is a high degree of specificity in the transfer.

This could have happened in all the experiments and it would still be "specific" as it is pulled by CD40. Even in the kinapse video, the cell seems to be attached to the old synapse site by a membrane tether that is seen stretching and most of the CD40 dissipates as the cell moves away. Thus, the majority of CD40L was still in the T cell membrane, not in ectosomes. Similarly, the proteomics captures a number of endocytic and ESCRT proteins that should remain in the cells when producing ectosomes, suggesting some membrane rupture occurred and the ligands in the SPLB pulled these complexes along. Working towards the mechanism, Figure 4 suggests that CD40L transfer to the SPLB beads is inhibited by MG132, but the inhibition is only partial and MGT132 has many effects beyond reducing free ubiquitin. In the same experiments the authors conclude that TCR transfer is not inhibited by MG132, yet there is a similar decrease in transfer as for the CD40L. This will need to be explained.

We have removed the MG132 experiments (previously Figure 4B and C) as the TSG101 and VPS4b CRISPR gene editing experiments provide a cleaner answer. Figure 4 is now focused only on the bystander transfer issue, which is biologically relevant.

2) What is the role of CD40 and ICOSL in mediating CD40L and ICOS release into ectosomes? The authors show that CD40L (or ICOS) accumulates in the synapse and in ectosomes strictly in correlation with the concentration of the CD40 (ICOSL) and independently of the concentration of antigen. Since it is possible that without the capture of CD40L by CD40, no accumulation would be seen simply because the CD40L is either internalised or secreted without capture, it is not clear what the steps are leading to inclusion of CD40L in the ectosomes and how does TCR regulate this process. Is it direct vesicle release, or plasma membrane incorporation followed by proteolytic cleavage or extrusion? In Figure 4, the authors show a depletion of CD40L in T cells that interacted with antigen and CD40, but not in cells that interacted only with antigen. However, it is not clear if the staining shows cell surface or total CD40L levels.

This is our current model: CD40L is inside the cell on ICAM-1 or ICAM-1 and CD40 PSLB and TCR triggering mobilizes it over the cell surface in small microclusters (subsection “CD40L is recruited to the IS and left by kinapses in a CD40 dependent manner” and new Figure 1C and D new Figure 1—figure supplement 1 and Video 2). CD40 in the PSLB appears to enable the CD40L microclusters to efficiently reach the cSMAC via well-documented F-actin flow (Video 1). Thus, CD40 helps the cell guide CD40L to the same compartments where TCR microclusters are forming and enables 50% of the resulting SE to have both TCR and CD40L microclusters. CD40L capture in vesicles seems to have a minimal requirement for CD40 in trans as a low density of 10/μm2 is sufficient, but there is some scaling of the CD40L transfer with UCHT1 density. In the context of proteolytic cleavage we have also inhibited both ADAM10 and ADAM17 with chemical inhibitors GI254023X and TAPI-2 respectively as well as specifically subjected the ADAM10 gene to CRISPR gene editing (new Figure 5—figure supplement 3). Rather than being reduced, we observe an increase in CD40L signal on the BSLB with the ADAM10 inhibitor or the ADAM10 editing. This suggests that CD40L is targeted by ADAM10, but this inhibits its accumulation in SE.

3) What is the role CD40L inclusion into ectosomes? It is known that CD40L works better if incorporated on particles or surfaces, however, this is already accomplished in the synapse. As the authors show, all the CD40L detectable in the synapse is captured via CD40, so CD40 signaling is already activated before ectosome release. After ectosome release, the ectosomes are bound to CD40 so they cannot amplify the signaling or spread to other APCs. Is the duration of CD40 signaling influenced?

The experiment we have done with dendritic cells in vitro reflects our model for why CD40L is captured in ectosomes – in order to be able to continue to have potent CD40 signalling in APCs after the T cell has left. That the ectosomes left on the substrate retain activity to signal dendritic cell maturation demonstrates in vitro that this form of CD40L acts independently of the T cell. Ours and prior experiments show that a similar amount of soluble CD40L would not have this activity (Figure 8D-F). More work is needed to understand the physiological import of this but we can demonstrate clearly in our system that the CD40L on the SE is active outside of the original IS. We present a working model in the last paragraph that the antigen specific Ca^2+^ signal we previously demonstrated for TCR enriched microvesicles may synergize with the CD40 signal we have demonstrated here for CD40L in SE to more fully activate APCs.

Reviewer #2:[…]1) Although the constellation of proteins enriched during the authors' BSLB-based purification strategy (e.g. Figure 2) does suggest that they are isolating SEs, I am not entirely convinced that the approach isn't just ripping off pieces of synaptic membrane indiscriminately. The Dustin lab has demonstrated quite nicely in the past that TSG101 and VPS4 are required for SE formation. They should demonstrate here that both proteins are required for CD40L and ICOS enrichment in their putative BSLB-based SE preparations.

As noted above, we have now used CRISPR gene editing in primary cells to eliminate ESCRT members including TSG101, VPS4b, CHMP4B, ALIX and EPN1 and found that CD40L transfer is indeed dependent on the ESCRT machinery. (subsection “Selective transfer of CD40L and ICOS into SE” and new Figure 2D, E and Figure 2—figure supplement 1B; subsection “Mass Spectrometry (MS) of SEs reveals enrichment of ESCRT proteins and TCR signaling.” and new Figure 5F and Figure 5—figure supplement 2)

2) In subsection “Mass Spectrometry (MS) of SEs reveals enrichment of ESCRT proteins and TCR signaling”, the authors write "We allowed 50 million CD4^+^ effector T cells to form IS on BSLB with ICAM-1, CD40, ICOSL in the presence or absence of UCHT1-Fab, disengaged the T cells and BSLB by incubation with ice-cold PBS/EDTA, sorted 5 million SE and subjected them to MS analysis after extraction of lipids." Do the authors mean "5 million BSLBs"? If they really do mean "SE", how did they dissociate SEs from BSLBs in their protocol?

It should be BSLB and we have corrected this (subsection “Mass Spectrometry (MS) of SEs reveals enrichment of ESCRT proteins and TCR signaling.”).

3) The polarization of CD40L shown in Figure 1C should be quantitated over multiple cells.

We have repeated the experiment and closer examination of the data revealed that there was a striking difference in CD40L location with and without TCR triggering, which we know highlight through additional examples and videos. We provide 3D videos that readers can navigate through and have also rescaled the live imaging of CD40L dynamics to emphasize how distinct CD40L and TCR microclusters converge on the cSMAC (subsection “CD40L is recruited to the IS and left by kinapses in a CD40 dependent manner” and new Figure 1C and D new Figure 1—figure supplement 1 and Video 2).

4) In subsection “Selective transfer of CD40L and ICOS into SE”, the authors write "This ICOSL driven TCR independent synapse may exert some control over migration of T cells, but it did not lead to CD40L transfer in any setting and thus does not appear to directly elicit delivery of T cell help." What is the basis for this statement? Did the authors measure antigen independent, ICOSL dependent help?

We agree to eliminate the statement "in any setting". We didn’t investigate ICOS dependent help per se, but ICAM-1, ICOS and CD40 is not sufficient for CD40L transfer. This is what we intended so we hope this is clearer.

5) In the Discussion section the authors write "It doesn't escape our attention that SE may provide an explanation for reports of antigen specific helper factors (43)." They should elaborate.

We have elaborated on this statement to include a description of the finding in the paper from Hodes lab and how the presence of TCR and CD40L on the SE could provide antigen specific help (Discussion paragraph three).

Reviewer #3:[…]A main question that is relevant and seems to be sidestepped throughout the manuscript, although somehow discussed, is how to differentiate between different kinds of extracellular vesicles that may form part of the T cell secretome released into the immune synapse. Authors may further address that the observed particles are only ectosomes and assess, or at least discuss, other sources for vesicles. A possible mechanism for ectosome formation may rely on the presence of particular proteins. Tetraspanins seem to be common, as well as some components of the ESCRT complexes. However, Epsin 1 protein, a well-known adaptor for AP2 and clathrin, but not known in the formation of intraluminal vesicles as components of multivesicular bodies (e.g.) is also there. If authors could demonstrate that the protein Epsin 1 is present at the plasma membrane, and contributes to the formation of ectosomes, and maybe not to the biogenesis of intraluminal vesicles, this would be a piece of evidence very supportive. The average size and number of detected TCR microclusters is similar to that of analyzed vesicles, which would support such a process.

We targeted EPN1 and ALIX using CRISPR gene editing. EPN1 is important for CD40L but not TCR transfer in SE (subsection “Mass Spectrometry (MS) of SEs reveals enrichment of ESCRT proteins and TCR signaling.” and new Figure 5F and Figure 5—figure supplement 2). Using Airyscan microscopy we also show that EPN1 proteins is present in the cSMAC formed by T cells on PSLB when coated with ICAM-1, CD40 and UCHT1-Fab (new Figure 5—figure supplement 1). ALIX loss has no effect on TCR transfer and minimal effect on CD40L. ALIX is in the correct place, but its function seems to be non-essential for transfer (new Figure 5F and Figure 5—figure supplement 2).